# SIMPLE POLICY OPTIMIZATION

## ABSTRACT

Model-free reinforcement learning algorithms have seen remarkable progress, but key challenges remain. Trust Region Policy Optimization (TRPO) is known for ensuring monotonic policy improvement through conservative updates within a trust region, backed by strong theoretical guarantees. However, its reliance on complex second-order optimization limits its practical efficiency. Proximal Policy Optimization (PPO) addresses this by simplifying TRPO's approach using ratio clipping, improving efficiency but sacrificing some theoretical robustness. This raises a natural question: Can we combine the strengths of both methods? In this paper, we introduce *Simple Policy Optimization* (SPO), a novel unconstrained first-order algorithm. SPO integrates the surrogate objective with Total Variation (TV) divergence instead of Kullback-Leibler (KL) divergence, achieving a balance between the theoretical rigor of TRPO and the efficiency of PPO. Our new objective improves upon ratio clipping, offering stronger theoretical properties and better constraining the probability ratio within the trust region. Empirical results demonstrate that SPO outperforms PPO with a simple implementation, particularly for training large, complex network architectures end-to-end.

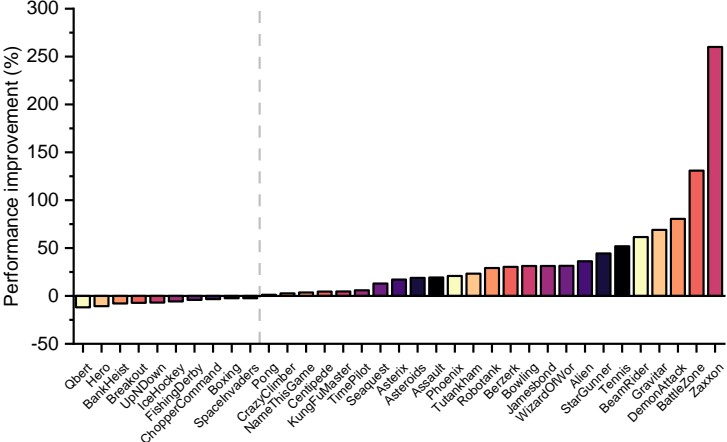

Figure 1: The performance of SPO compared to PPO across 35 games in Atari 2600.

## 1 INTRODUCTION

Deep Reinforcement Learning (DRL) has achieved great success in recent years, notably in board games (Silver et al., 2016; 2017; 2018), video games (Mnih et al., 2015; Berner et al., 2019; Vinyals et al., 2019; Ye et al., 2020), autonomous driving (Kiran et al., 2021; Zhu & Zhao, 2021; Teng et al., 2023), and robotics (Makoviychuk et al., 2021; Brunke et al., 2022; Han et al., 2023). Policy gradient (PG) methods (Sutton & Barto, 2018; Lehmann, 2024), as a major paradigm in RL, have been widely adopted by the academic community. One main practical challenge of PG methods is to reduce the variance of the gradients while keeping the bias low (Sutton et al., 2000; Schulman et al., 2015b). In this context, a widely used technique is to add a baseline (usually the value function) when sampling an estimate of the action-value function (Greensmith et al., 2004). Another challenge of PG methods

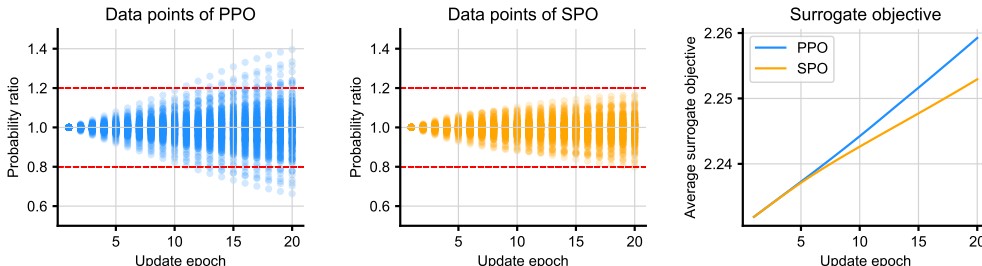

Figure 2: Optimization behavior of PPO (left) and SPO (middle). The optimization process of PPO and SPO in the Hopper-v4 environment is visualized, with the red line representing the probability ratio bound. Although PPO achieves better optimization of the surrogate objective, it does so by breaking the ratio constraints. In contrast, SPO optimizes the objective more effectively while adhering to the constraint.

is to estimate the proper step size for the policy update (Kakade & Langford, 2002; Schulman et al., 2015a). Given that the training data strongly depends on the current policy, a large step size may result in a collapse of policy performance, whereas a small one may impair the sample efficiency of the algorithm.

To address these challenges, in Trust Region Policy Optimization (TRPO), Schulman et al. (2015a) proved that minimizing a certain surrogate objective guarantees policy improvement with non-trivial step sizes. Subsequently, the TRPO algorithm was derived through a series of approximations, which impose a trust region constraint during the policy iterations, leading to monotonic policy improvement in theory. However, given the complexity of second-order optimization, TRPO is highly inefficient and can be hard to extend to large-scale RL environments. Proximal Policy Optimization (PPO) (Schulman et al., 2017) is designed to enforce comparable constraints on the difference between successive policies during the training process, while only using first-order optimization. By clipping the current data that exceeds the probability ratio limit to a constant, PPO attempts to remove the high incentive for pushing the current policy away from the old one. It has been shown that PPO can be effectively extended to large-scale complex control tasks (Ye et al., 2020; Makoviychuk et al., 2021).

Despite its success, the optimization behavior of PPO remains insufficiently understood. Although PPO aims to constrain the probability ratio deviations between successive policies, it often fails to keep these ratios within bounds (Ilyas et al., 2018; Engstrom et al., 2020). In some tasks, the ratios can even escalate to values as high as $40$[1] (Wang et al., 2020). Furthermore, studies have revealed that PPO's performance is highly dependent on "code-level optimizations" (Andrychowicz et al., 2021). The implementation of PPO includes numerous code-level details that critically influence its effectiveness (Engstrom et al., 2020; Huang et al., 2022a;b).

In this paper, we propose a new algorithm named Simple Policy Optimization (SPO) designed to more effectively bound probability ratios through a novel objective function. The differences in optimization behavior between PPO and SPO are illustrated in Fig. 2. Our main contributions are as follows:

- We theoretically prove that optimizing a tighter performance lower bound using Total Variation (TV) divergence constrained space results in more consistent policy improvement.

- To overcome PPO's limitation in constraining probability ratios, we propose a new objective function, leading to the development of the proposed SPO algorithm.

- Experiments benchmark various policy gradient algorithms across different environments, showing that SPO can achieve competitive performance with a simple implementation, improved sample efficiency, and easier training of deeper policy networks.

---

[1]The clipping parameter $\epsilon$ is set to $0.2$, so the upper clipping range is $1.2$.

## 2 PRELIMINARIES

### 2.1 REINFORCEMENT LEARNING

Online reinforcement learning is a mathematical framework for sequential decision-making, which is generally defined by the Markov Decision Process (MDP) $\mathcal{M} = (\mathcal{S}, \mathcal{A}, r, \mathcal{P}, \rho_0, \gamma)$, where $\mathcal{S}$ and $\mathcal{A}$ represent the state space and action space, $r : \mathcal{S} \times \mathcal{A} \mapsto \mathbb{R}$ is the reward function, $\mathcal{P} : \mathcal{S} \times \mathcal{A} \times \mathcal{S} \mapsto [0, 1]$ is the probability distribution of the state transition function, $\rho_0 : \mathcal{S} \mapsto [0, 1]$ is the initial state distribution, while $\gamma \in (0, 1]$ is the discount factor.

Suppose that an agent interacts with the environment following policy $\pi$, i.e., $a \sim \pi(\cdot|s)$ and obtains a trajectory $\tau = (s_0, a_0, r_0, \ldots, s_t, a_t, r_t, \ldots)$, where $r_t = r(s_t, a_t)$. The goal of RL is to learn a policy that maximizes the expected return $\mathbb{E}_{\tau \sim \pi} \left[ \sum_{t=0}^{\infty} \gamma^t r_t \right]$[2]. The state-value function and value function are defined as

$$Q_\pi(s_t, a_t) = \mathbb{E}_{s_{t+1}, a_{t+1}, \ldots} \left[ \sum_{k=0}^{\infty} \gamma^k r(s_{t+k}, a_{t+k}) \right], \quad V_\pi(s_t) = \mathbb{E}_{a_t \sim \pi(\cdot|s_t)} \left[ Q_\pi(s_t, a_t) \right]. \quad (1)$$

Given $Q_\pi$ and $V_\pi$, the advantage function can be expressed as $A_\pi(s_t, a_t) = Q_\pi(s_t, a_t) - V_\pi(s_t)$.

### 2.2 TRUST REGION POLICY OPTIMIZATION

Classic policy gradient methods cannot reuse data and are highly sensitive to the hyperparameters. To address these issues, in Trust Region Policy Optimization (TRPO), Schulman et al. (2015a) derived a lower bound for policy improvement. Before that, Kakade & Langford (2002) first proved the following policy performance difference theorem:

**Theorem 2.1.** *(Kakade & Langford, 2002) Let $\mathbb{P}(s_t = s|\pi)$ represents the probability of the $t$-th state equals to $s$ in trajectories generated by the agent following policy $\pi$, and $\rho_\pi(s) = \sum_{t=0}^{\infty} \gamma^t \mathbb{P}(s_t = s|\pi)$ represents the unnormalized discounted visitation frequencies. Given any two policies, $\pi$ and $\tilde{\pi}$, their performance difference can be measured by*

$$\eta(\tilde{\pi}) - \eta(\pi) = \mathbb{E}_{s \sim \rho_{\tilde{\pi}}(\cdot), a \sim \tilde{\pi}(\cdot|s)} \left[ A_\pi(s, a) \right] = \sum_s \rho_{\tilde{\pi}}(s) \sum_a \tilde{\pi}(a|s) \cdot A_\pi(s, a), \quad (2)$$

*where $\eta(\pi) = \mathbb{E}_{\tau \sim \pi} \left[ \sum_{t=0}^{\infty} \gamma^t r(s_t, a_t) \right]$. The notation $\mathbb{E}_{\tau \sim \pi}$ indicates the expected return of the trajectory $\tau$ generated by the agent following policy $\pi$, i.e., $s_0 \sim \rho_0(\cdot), a_t \sim \pi(\cdot|s_t), r_t \sim r(s_t, a_t), s_{t+1} \sim \mathcal{P}(\cdot|s_t, a_t)$.*

The proof of Theorem 2.1 see Appendix F. The key intuition is that the new policy $\tilde{\pi}$ will improve (or remain constant) as long as it has a nonnegative expected advantage at every state $s$. Then, define $L_\pi(\tilde{\pi}) = \eta(\pi) + \mathbb{E}_{s \sim \rho_\pi(\cdot), a \sim \tilde{\pi}(\cdot|s)} \left[ A_\pi(s, a) \right]$, the following performance lower bound is given:

**Theorem 2.2.** *(Schulman et al., 2015a) Given any two policies, $\pi$ and $\tilde{\pi}$, the following bound holds:*

$$\eta(\tilde{\pi}) \geq L_\pi(\tilde{\pi}) - \frac{4\epsilon\gamma}{(1-\gamma)^2} \cdot D_{\text{TV}}^{\max}(\pi, \tilde{\pi})^2, \quad (3)$$

*where $\epsilon = \max_{s,a} |A_\pi(s, a)|$, $D_{\text{TV}}(p, q) = \frac{1}{2} \sum |p_i - q_i|$ is the Total Variation (TV) divergence and $D_{\text{TV}}^{\max}$ means $\max_s D_{\text{TV}}$.*

Given this lower bound, we can actually achieve the monotonic improvement of policy performance. According to the inequality between the TV divergence and the KL divergence $D_{\text{TV}}(\pi, \tilde{\pi})^2 \leq D_{\text{KL}}(\pi, \tilde{\pi})$, we have $\eta(\tilde{\pi}) \geq L_\pi(\tilde{\pi}) - \frac{4\epsilon\gamma}{(1-\gamma)^2} \cdot D_{\text{KL}}^{\max}(\pi, \tilde{\pi})$. Then, define $M_\pi(\tilde{\pi}) = L_\pi(\tilde{\pi}) - C \cdot D_{\text{KL}}^{\max}(\pi, \tilde{\pi})$, where $C = 4\epsilon\gamma/(1-\gamma)^2$. It follows that $\eta(\tilde{\pi}) \geq M_\pi(\tilde{\pi})$ and $\eta(\pi) = M_\pi(\pi)$.

Consequently, the difference $\eta(\tilde{\pi}) - \eta(\pi)$ is bounded below by $M_\pi(\tilde{\pi}) - M_\pi(\pi)$. This implies that the original objective $\eta$ can be optimized indirectly by maximizing the lower bound $M_\pi$. Specifically,

$$\eta(\tilde{\pi}) - \eta(\pi) \geq M_\pi(\tilde{\pi}) - M_\pi(\pi) = \mathbb{E}_{s \sim \rho_\pi(\cdot), a \sim \pi(\cdot|s)} \left[ \frac{\tilde{\pi}(a|s)}{\pi(a|s)} \cdot A_\pi(s, a) \right] - C \cdot D_{\text{KL}}^{\max}(\pi, \tilde{\pi}). \quad (4)$$

---

[2]Note that we are discussing an infinite-horizon Markov Decision Process (MDP) here. For a finite MDP of length $T$, all reward values beyond $T$ are set to zero.

At this point, the subscripts of the expectation in Eq. (2) are replaced from $s \sim \rho_{\tilde{\pi}}(\cdot)$ and $a \sim \tilde{\pi}(\cdot|s)$ to $s \sim \rho_{\pi}(\cdot)$ and $a \sim \pi(\cdot|s)$, which means that we can now reuse the current data. Through a heuristic approximation, the maximum KL divergence $D_{\mathrm{KL}}^{\max}$ in Eq. (4) is approximated as the average KL divergence and then used as a constraint to obtain the formalization of TRPO:

$$\max_{\theta} \ \mathbb{E}_{(s_t, a_t) \sim \pi_{\theta_{\mathrm{old}}}} \left[ \frac{\pi_\theta(a_t|s_t)}{\pi_{\theta_{\mathrm{old}}}(a_t|s_t)} \cdot \hat{A}(s_t, a_t) \right], \tag{5}$$
$$\text{s.t.} \ \ \mathbb{E}\left[ D_{\mathrm{KL}}(\pi_{\theta_{\mathrm{old}}}, \pi_\theta) \right] \le \delta.$$

This problem includes a constraint where $\delta$ is a hyperparameter that limits the KL divergence between successive policies, typically set to 0.01. $\hat{A}(s_t, a_t)$ is an estimate of the advantage function, and the objective is called "surrogate objective".

## 2.3 PROXIMAL POLICY OPTIMIZATION

Due to the necessity of solving a constrained optimization problem (5) in each update, TRPO is highly inefficient and can be challenging to apply to large-scale reinforcement learning tasks.

Schulman et al. (2017) proposed a new objective called "clipped surrogate objective", in which the algorithm is named Proximal Policy Optimization (PPO). PPO retains some of the benefits of TRPO but is much easier to implement and involves only unconstrained first-order optimization.

The "clipped surrogate objective", also called PPO-Clip, adopts a ratio clipping function, which is

$$J_{\mathrm{clip}}(\theta) = \mathop{\mathbb{E}}_{(s_t, a_t) \sim \pi_{\theta_{\mathrm{old}}}} \left\{ \min \left[ r_t(\theta) \cdot \hat{A}(s_t, a_t), \mathrm{clip}\left( r_t(\theta), 1 - \epsilon, 1 + \epsilon \right) \cdot \hat{A}(s_t, a_t) \right] \right\}, \tag{6}$$

where

$$r_t(\theta) = \frac{\pi_\theta(a_t|s_t)}{\pi_{\theta_{\mathrm{old}}}(a_t|s_t)}, \ \ \mathrm{clip}\left( r_t(\theta), 1 - \epsilon, 1 + \epsilon \right) = \begin{cases} 1 - \epsilon, & r_t(\theta) < 1 - \epsilon; \\ r_t(\theta), & 1 - \epsilon \le r_t(\theta) \le 1 + \epsilon; \\ 1 + \epsilon, & r_t(\theta) > 1 + \epsilon. \end{cases} \tag{7}$$

The gradient of PPO-Clip, given the training data $(s_t, a_t)$, can be expressed as follows:

$$\nabla_\theta J_{\mathrm{clip}}(\theta) = \begin{cases} \nabla_\theta r_t(\theta) \cdot \hat{A}(s_t, a_t), & \hat{A}(s_t, a_t) > 0, r_t(\theta) \le 1 + \epsilon; \\ \nabla_\theta r_t(\theta) \cdot \hat{A}(s_t, a_t), & \hat{A}(s_t, a_t) < 0, r_t(\theta) \ge 1 - \epsilon; \\ 0, & \text{otherwise.} \end{cases} \tag{8}$$

In other words, PPO-Clip aims to remove the high incentive for pushing the current policy away from the old one. PPO-Clip has gained wide adoption in the academic community due to its simplicity and superior performance.

## 3 METHODOLOGY

PPO attempts to limit the differences between successive policies through ratio clipping. However, Wang et al. (2020) proved the following theorem:

**Theorem 3.1.** *(Wang et al., 2020) Assume that for discrete action space tasks where $|\mathcal{A}| \ge 3$ or continuous action space tasks where the output of the policy $\pi_\theta$ follows a multivariate Gaussian distribution. Let $\Theta = \{\theta | 1 - \epsilon \le r_t(\theta) \le 1 + \epsilon\}$, we have $\sup_{\theta \in \Theta} D_{\mathrm{KL}}(\pi_{\theta_{\mathrm{old}}}(\cdot|s_t), \pi_\theta(\cdot|s_t)) = +\infty$ for both discrete and continuous action space tasks.*

Theorem 3.1 demonstrates that the KL divergence $D_{\mathrm{KL}}(\pi_{\theta_{\mathrm{old}}}(\cdot|s_t), \pi_\theta(\cdot|s_t))$ is not necessarily bounded even if the probability ratio $r_t(\theta)$ is bounded. However, this theorem considers only an extreme case involving a single data point, which is less typical than the batch processing used in training data. On a broader scale, the clipped ratio strategy employed by Proximal Policy Optimization (PPO) aims to bound the total variation (TV) divergence for sufficient batch sizes. This relationship is formalized as $D_{\mathrm{TV}}^{\max}(\pi, \tilde{\pi})^2 = \frac{1}{4} \max_s \left[ \mathbb{E}_{a \sim \pi(\cdot|s)} \left| \frac{\tilde{\pi}(a|s)}{\pi(a|s)} - 1 \right| \right]^2$. Details on how this

bounding strategy relates to the probability ratio are provided in Appendix G. Furthermore, using this approach, the performance lower bound (3) can be expressed as

$$\eta(\tilde{\pi}) \geq L_\pi(\tilde{\pi}) - \frac{4\epsilon\gamma}{(1-\gamma)^2} \cdot \frac{1}{4} \max_s \left[ \mathbb{E}_{a \sim \pi(\cdot|s)} \left| \frac{\tilde{\pi}(a|s)}{\pi(a|s)} - 1 \right| \right]^2. \tag{9}$$

Similar to (4), the performance difference lower bound of PPO can be derived (Queeney et al., 2021):

$$\eta(\tilde{\pi}) - \eta(\pi) \geq \mathbb{E}_{s \sim \rho_\pi(\cdot), a \sim \pi(\cdot|s)} \left[ \frac{\tilde{\pi}(a|s)}{\pi(a|s)} \cdot A_\pi(s,a) \right] - \frac{C}{4} \cdot \max_s \left[ \mathbb{E}_{a \sim \pi(\cdot|s)} \left| \frac{\tilde{\pi}(a|s)}{\pi(a|s)} - 1 \right| \right]^2. \tag{10}$$

Finally, we also found that PPO, which aims to bound the TV divergence, can offer a larger solution space compared to methods that incorporate a looser KL divergence as a constraint (e.g., TRPO). To illustrate this, we present the following proposition:

**Proposition 3.2.** *Given any state $s \in \mathcal{S}$ and the old policy $\pi$, define the solution spaces under the TV and KL divergence constraints as follows:*

$$\Omega_{\mathrm{TV}} = \{\tilde{\pi} \mid D_{\mathrm{TV}} \left[ \pi(\cdot|s), \tilde{\pi}(\cdot|s) \right]^2 \leq \delta, \forall s \in \mathcal{S}\}, \ \Omega_{\mathrm{KL}} = \{\tilde{\pi} \mid D_{\mathrm{KL}} \left[ \pi(\cdot|s), \tilde{\pi}(\cdot|s) \right] \leq \delta, \forall s \in \mathcal{S}\}, \tag{11}$$

*where $\delta$ is a predefined threshold. Under these constraints, we establish that $\Omega_{\mathrm{KL}} \subset \Omega_{\mathrm{TV}}$.*

*Proof.* For any $s \in \mathcal{S}$ and $\tilde{\pi} \in \Omega_{\mathrm{KL}}$, using Pinsker's inequality, we have $D_{\mathrm{TV}} \left[ \pi(\cdot|s), \tilde{\pi}(\cdot|s) \right]^2 \leq D_{\mathrm{KL}} \left[ \pi(\cdot|s), \tilde{\pi}(\cdot|s) \right] \leq \delta$, therefore $\tilde{\pi} \in \Omega_{\mathrm{TV}}$, which means $\Omega_{\mathrm{KL}} \subset \Omega_{\mathrm{TV}}$, concluding the proof. $\square$

Additionally, the optimal solution to the performance difference lower bound in the TV divergence-constrained space, $\Omega_{\mathrm{TV}}$, is expected to be superior. We now present the following theorem:

**Theorem 3.3.** *Given the old policy $\pi$, and $\Omega_{\mathrm{TV}}, \Omega_{\mathrm{KL}}$ presented in Proposition 3.2, let*

$$\mathcal{L}_\pi^{\mathrm{TV}}(\tilde{\pi}) = L_\pi(\tilde{\pi}) - L_\pi(\pi) - C \cdot D_{\mathrm{TV}}^{\max}(\pi, \tilde{\pi})^2, \ \mathcal{L}_\pi^{\mathrm{KL}}(\tilde{\pi}) = L_\pi(\tilde{\pi}) - L_\pi(\pi) - C \cdot D_{\mathrm{KL}}^{\max}(\pi, \tilde{\pi}) \tag{12}$$

*be the lower bounds of performance improvement with TV divergence and KL divergence. Denote $\tilde{\pi}_{\mathrm{TV}}^* = \arg\max_{\tilde{\pi} \in \Omega_{\mathrm{TV}}} \mathcal{L}_\pi^{\mathrm{TV}}(\tilde{\pi})$ and $\tilde{\pi}_{\mathrm{KL}}^* = \arg\max_{\tilde{\pi} \in \Omega_{\mathrm{KL}}} \mathcal{L}_\pi^{\mathrm{KL}}(\tilde{\pi})$, then $\mathcal{L}_\pi^{\mathrm{TV}}(\tilde{\pi}_{\mathrm{TV}}^*) \geq \mathcal{L}_\pi^{\mathrm{KL}}(\tilde{\pi}_{\mathrm{KL}}^*)$.*

*Proof.* Since $\Omega_{\mathrm{KL}} \subset \Omega_{\mathrm{TV}}$, we have

$$\begin{aligned} \mathcal{L}_\pi^{\mathrm{TV}}(\tilde{\pi}_{\mathrm{TV}}^*) &\geq \mathcal{L}_\pi^{\mathrm{TV}}(\tilde{\pi}_{\mathrm{KL}}^*) = L_\pi(\tilde{\pi}_{\mathrm{KL}}^*) - L_\pi(\pi) - C \cdot D_{\mathrm{TV}}^{\max}(\pi, \tilde{\pi}_{\mathrm{KL}}^*)^2 \\ &\geq L_\pi(\tilde{\pi}_{\mathrm{KL}}^*) - L_\pi(\pi) - C \cdot D_{\mathrm{KL}}^{\max}(\pi, \tilde{\pi}_{\mathrm{KL}}^*) = \mathcal{L}_\pi^{\mathrm{KL}}(\tilde{\pi}_{\mathrm{KL}}^*), \end{aligned} \tag{13}$$

concluding the proof. $\square$

Based on the Proposition 3.2 and Theorem 3.3, we have the following conclusion:

> **Conclusion**
>
> Optimizing the lower bound with TV divergence constrains offers a more effective solution space than using KL divergence constrains, leading to better policy improvement.

As a result, we aim to solve the following constrained optimization problem:

$$\begin{aligned} \max_\theta \ & \mathbb{E}_{(s_t, a_t) \sim \pi_{\theta_{\mathrm{old}}}} \left[ \frac{\pi_\theta(a_t|s_t)}{\pi_{\theta_{\mathrm{old}}}(a_t|s_t)} \cdot \hat{A}(s_t, a_t) \right], \\ \mathrm{s.t.} \ & \left| \frac{\pi_\theta(a_t|s_t)}{\pi_{\theta_{\mathrm{old}}}(a_t|s_t)} - 1 \right| \leq \epsilon, \ \forall (s_t, a_t) \sim \pi_{\theta_{\mathrm{old}}}. \end{aligned} \tag{14}$$

Notably, similar forms have already been obtained in previous work (Vuong et al., 2018). PPO attempts to satisfy the constraints of (14) through ratio clipping, but this does not prevent excessive ratio deviations (see Fig. 2). The underlying reason, however, is that ratio clipping causes certain data points to stop contributing to the gradient updates according to (8). Over multiple iterations,

---

**Algorithm 1** Simple Policy Optimization (SPO)

---

1: **Initialize:** Policy network $\pi_\theta$, value network $V_\phi$, probability ratio deviation hyperparameter $\epsilon$, value loss coefficient $c_1$, policy entropy coefficient $c_2$
2: **Output:** Optimal policy network $\pi_{\theta*}$
3: **while** not converged **do**
        # Data collection
4:      Collect data $X = \{(s_i, a_i, r_i)\}_{i=1}^N$ using the current policy network $\pi_\theta$
        # The networks before updating
5:      $\pi_{\theta_{\text{old}}} \leftarrow \pi_\theta, \ V_{\phi_{\text{old}}} \leftarrow V_\phi$
        # Estimate the advantage $\hat{A}(s_t, a_t)$ based on $V_{\phi_{\text{old}}}$
6:      Use GAE (Schulman et al., 2015b) technique to estimate the advantage $\hat{A}(s_t, a_t)$
        # Estimate the return $\hat{R}_t$
7:      $\hat{R}_t \leftarrow V_{\phi_{\text{old}}}(s_t) + \hat{A}(s_t, a_t)$
8:      **for** each training epoch **do**
            # Compute policy loss $\mathcal{L}_p$ (This is the only difference between SPO and PPO)
9:          $\mathcal{L}_p \leftarrow - \left\{ r_t(\theta) \cdot \hat{A}(s_t, a_t) - \frac{|\hat{A}(s_t, a_t)|}{2\epsilon} \cdot [r_t(\theta) - 1]^2 \right\}$
            # Compute policy entropy $\mathcal{L}_e$ and value loss $\mathcal{L}_v$
10:         $\mathcal{L}_e \leftarrow \mathcal{H}(\pi_\theta(\cdot|s_t)), \ \mathcal{L}_v \leftarrow \frac{1}{2}[V_\phi(s_t) - \hat{R}_t]^2$
            # Compute total loss $\mathcal{L}$
11:         $\mathcal{L} \leftarrow \mathcal{L}_p + c_1 \mathcal{L}_v - c_2 \mathcal{L}_e$
            # Update parameters $\theta$ and $\phi$ through backpropagation, $\lambda_\theta$ and $\lambda_\phi$ is the step sizes
12:         $\theta \leftarrow \theta - \lambda_\theta \nabla_\theta \mathcal{L}, \ \phi \leftarrow \phi - \lambda_\phi \nabla_\phi \mathcal{L}$
13:     **end for**
14: **end while**

---

this can lead to uncontrollable updates, as the absence of corrective gradients prevents the policy from recovering. To overcome this issue with ratio clipping, we propose the following objective:

$$J(\theta) = \mathop{\mathbb{E}}_{(s_t, a_t) \sim \pi_{\theta_{\text{old}}}} \left\{ r_t(\theta) \cdot \hat{A}(s_t, a_t) - \frac{|\hat{A}(s_t, a_t)|}{2\epsilon} \cdot [r_t(\theta) - 1]^2 \right\}. \tag{15}$$

The details of the objective $J(\theta)$ will be discussed in the following section, and the pseudo-code is shown in Algorithm 1.

## 4 THEORETICAL RESULTS

In this section, we provide some theoretical analysis of the differences between PPO and SPO, demonstrate that SPO can be more effective in constraining probability ratios during training.

### 4.1 OBJECTIVE CLASS

We consider the class of objectives and present the following definition:

**Definition 4.1** ($\epsilon$-aligned). We say that the objective function $f(r, A, \epsilon)$ is $\epsilon$-aligned, if $f$ involves $rA$ term, and for any given $\bar{A} \neq 0$ and $\bar{\epsilon} > 0$, the function $g(r) = f(r, \bar{A}, \bar{\epsilon})$ is convex and attains its maximum value at $r = 1 + \text{sign}(\bar{A}) \cdot \bar{\epsilon}$, where $\text{sign}(\cdot)$ is the sign function.

The objective of PPO in (6) can be expressed as

$$f_{\text{ppo}} = \min\left[rA, \text{clip}(r, 1 - \epsilon, 1 + \epsilon)A\right]. \tag{16}$$

It can be immediately obtained that $f_{\text{ppo}}$ is not $\epsilon$-aligned, as $f_{\text{ppo}}$ does not contain the term $rA$ in some cases according to (7). Next, we will demonstrate several desirable properties of the function $f$ that satisfies the $\epsilon$-aligned condition.

**Theorem 4.2.** *Given an objective function $f(r, A, \epsilon)$, which is $\epsilon$-aligned. If $f$ is also differentiable with respect to $r$, then for any given $\bar{A} \neq 0, \bar{\epsilon} > 0$, and any initial $r \in (0, +\infty)$, the gradient ascent*

*algorithm $r \leftarrow r + \nabla_r f$ will drive $f$ to converge to the optimal solution $r^* = \arg\max_r f(r, \bar{A}, \bar{\epsilon})$, which is also the optimal solution to the constrained optimization problem with ratio bound:*

$$\max_r \ r\bar{A}, \ \text{s.t.} \ |r - 1| \leq \bar{\epsilon}. \tag{17}$$

*Proof.* Given that $f$ is differentiable with respect to $r$ and $f$ is $\epsilon$-aligned, it follows that the function $g(r) = f(r, \bar{A}, \bar{\epsilon})$ is convex and differentiable with respect to $r$, enabling it to converge to the optimal solution $r^* = 1 + \text{sign}(\bar{A}) \cdot \bar{\epsilon}$. Then, for the constrained optimization problem (17), the objective is linear, so the optimal solution is

$$\tilde{r}^* = \begin{cases} 1 - \bar{\epsilon}, & \bar{A} < 0; \\ 1 + \bar{\epsilon}, & \bar{A} > 0, \end{cases} \tag{18}$$

which means $\tilde{r}^* = 1 + \text{sign}(\bar{A}) \cdot \bar{\epsilon} = r^*$, concluding the proof. $\square$

Theorem 4.2 demonstrates that if the objective $f$ satisfies the $\epsilon$-aligned property, while also being differentiable with respect to $r$, then it is capable of converging to an optimal solution that is consistent with the solution of directly solving the constrained problem with ratio bound, while only using a first-order optimizer (e.g., Adam).

Now, the objective of SPO in (15) can be expressed as

$$f_{\text{spo}} = rA - \frac{|A|}{2\epsilon}(r - 1)^2. \tag{19}$$

For $f_{\text{spo}}$, we present the following theorem:

**Theorem 4.3.** $f_{\text{spo}}$ *is $\epsilon$-aligned and also differentiable with respect to $r$.*

*Proof.* Obviously, $f_{\text{spo}}$ is differentiable with respect to $r$ since $f_{\text{spo}}$ is a quadratic polynomial of $r$, which implies that $g(r) = f(r, \bar{A}, \bar{\epsilon})$ is also convex. Let $\partial f_{\text{spo}}(r, A, \epsilon)/\partial r = 0$, we have

$$\frac{\partial f_{\text{spo}}(r, A, \epsilon)}{\partial r} = A - \frac{|A|}{\epsilon}(r - 1) = 0, \tag{20}$$

thus $r^* = 1 + \text{sign}(A) \cdot \epsilon$ is the optimal solution for $f_{\text{spo}}$, concluding the proof. $\square$

### 4.2 ANALYSIS OF NEW OBJECTIVE

We show that the optimization process of SPO can more effectively bound the probability ratio, as can be seen from Fig. 3. The largest circular area in the figure represents the boundary on the probability ratio. The green circles represent data points with non-zero gradients during the training process, while the gray circles represent data points with zero gradients.

During the training process of PPO, certain data points that exceed the probability ratio bound cease to provide gradients. In contrast, all data points in SPO contribute gradients that guide the optimization towards the constraint boundary. As training progresses, PPO will accumulate more gray circles that no longer provide gradients and may be influenced by the harmful gradients from

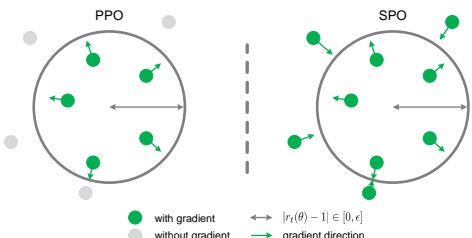

Figure 3: In PPO, certain data points exhibit zero gradients, while in SPO, all data points generate non-zero gradients that guide towards the constraint boundary.

green circles. This phenomenon could potentially push the gray circles further away from the constraint boundary. In contrast, the gradient directions of all data points in SPO point towards the constraint boundary. This indicates that SPO imposes stronger constraints on the probability ratio.

## 5 EXPERIMENTS

We report results on the Atari 2600 (Bellemare et al., 2013; Machado et al., 2018) and MuJoCo (Todorov et al., 2012) benchmarks. In all our experiments, we utilize the RL library Gymnasium (Towers et al., 2024), which serves as a central abstraction to ensure broad interoperability between benchmark environments and training algorithms.

### 5.1 COMPARING ALGORITHMS

Our implementation of SPO is compared against PPO-Clip (Schulman et al., 2017), PPO-Penalty (Schulman et al., 2017), SPU (Vuong et al., 2018), PPO-RB (Wang et al., 2020), TR-PPO (Wang et al., 2020), TR-PPO-RB (Wang et al., 2020), and RPO (Gan et al., 2024) in the MuJoCo benchmark. For all algorithms, agents collect experience from 8 parallel worker copies into a single buffer, with each environment running for 256 steps before the next training session.

In all experiments, we use the hyperparameters provided in the Appendix D unless otherwise specified. For the fairness of the experiments, we referred to the original paper for hyperparameters specific to the algorithm, otherwise keep them consistent. We compute the algorithm's performance across ten separate runs with different random seeds. In addition, we emphasize that in all comparative experiments involving the same settings for SPO and PPO, the only modification in SPO is replacing the PPO's objective with Eq. (15), highlighting the simplicity and efficiency of SPO.

Due to the absence of human score baselines in the Mujoco (Todorov et al., 2012), we normalize the algorithms' performance across all environments using the training data of PPO-Clip, specifically,

$$\text{normalized(score)} = \frac{\text{score} - \min}{\max - \min}, \tag{21}$$

where max and min represent the maximum and minimum validation returns of PPO-Clip during training, respectively.

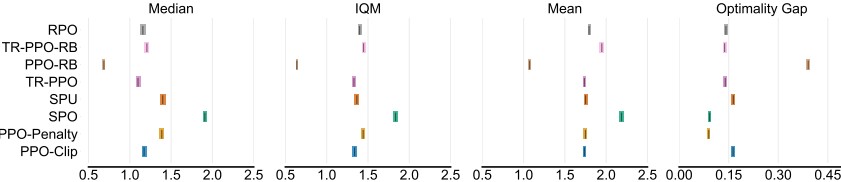

Figure 4: Aggregate metrics on Mujoco-v4 with 95% CIs based on 6 environments. We collected the validation returns of each algorithm over the last 1% training steps across ten random seeds. In this context, higher median, IQM and mean scores and lower optimality gap are better.

As suggested in Agarwal et al. (2021), we employ stratified bootstrap confidence intervals to assess the confidence intervals of the algorithm and evaluate the composite metrics of SPO against other baselines, as illustrated in Figure 4. It can be observed that SPO achieved the best performance across nearly all statistical metrics, which fully demonstrates the strong potential of SPO.

For the Atari 2600 benchmark (Bellemare et al., 2013), the main results are presented in the Appendix H and Fig. 1. Although SPO did not consistently outperform PPO under the same settings, we will demonstrate in the following section that this is primarily due to the limited expressive capacity of the network and the overly restrictive ratio constraints imposed by SPO.

### 5.2 SCALING POLICY NETWORK

The high performance of the PPO algorithm often relies on a carefully designed model architecture (Huang et al., 2022a). To investigate how scaling policy network size impacts the sample efficiency of both PPO and SPO in MuJoCo, the number of policy network layers was increased without altering the hyperparameters or other settings. The standard deviation of the algorithm's performance was computed and visualized across five separate runs with different random seeds. The results, shown in Fig. 5, 6 and Tab. 5, where the ratio deviation indicates the largest value of average ratio deviation in a batch, i.e., $\frac{1}{|\mathcal{D}|} \sum_{(s_t, a_t) \sim \mathcal{D}} |r_t(\theta) - 1|$, which should typically be less than $\epsilon = 0.2$.

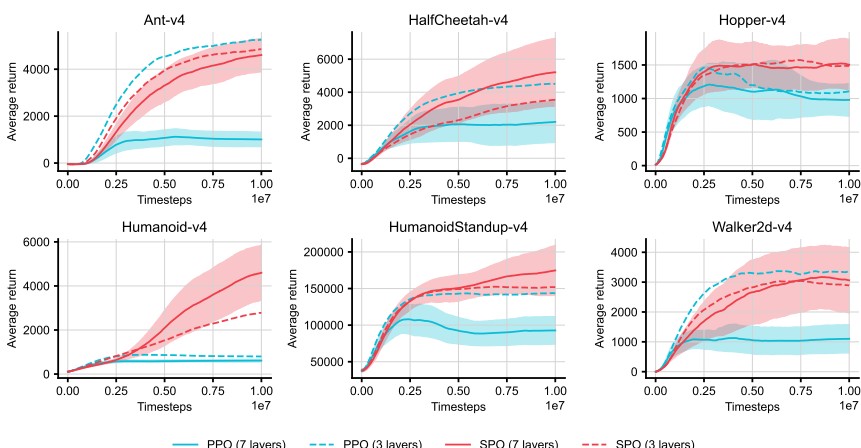

Figure 5: Training performance of PPO and SPO with different policy network layers in MuJoCo benchmark. The mean and standard deviation are shown across five random seeds.

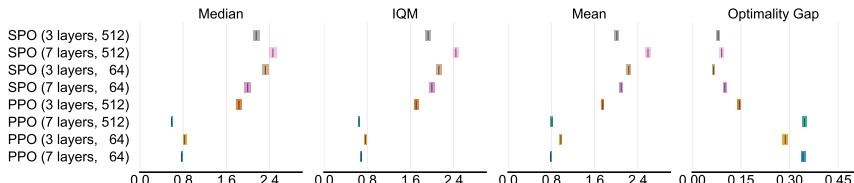

Figure 6: Aggregate metrics on Mujoco-v4 with 95% CIs based on 6 environments, comparing PPO and SPO with different policy network layers and mini-batch size using PPO-normalized score.

It can be observed that as the network deepens, the performance of PPO collapses in most environments, with uncontrollable probability ratio deviations. In contrast, the performance of SPO outperforms that of shallow networks in almost all environments and constrains the probability ratio deviation effectively. Furthermore, the statistical metrics of SPO generally outperform PPO's and demonstrate relative robustness to variations in network depth and mini-batch size.

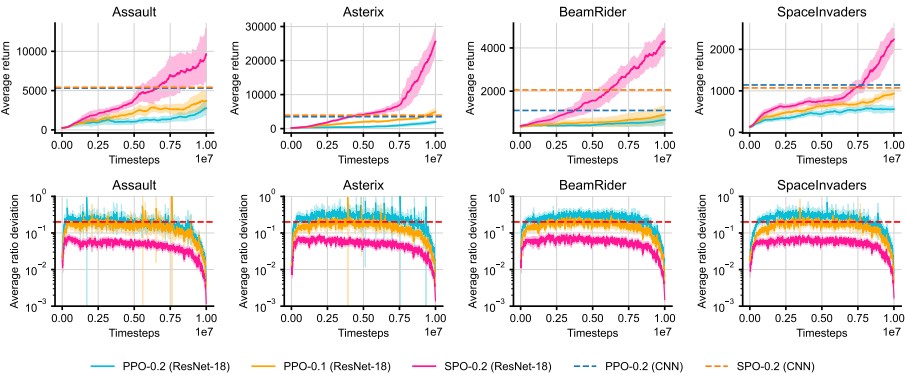

Figure 7: Training curves of SPO using ResNet-18 as the encoder compared to the PPO and SPO with original CNN. The mean and standard deviation are shown across three random seeds.

We also trained the ResNet-18[3] as the encoder on the Atari 2600 benchmark, the results are shown in Fig. 7. As the network's capacity increases, the performance of SPO is significantly improved,

---

[3]Since Bjorck et al. (2021) demonstrated that batch normalization is harmful to RL, we removed batch normalization and adjusted ResNet-18 for input and output from our implementation.

particularly in the Asterix environment, where the final reward is almost *seven* times the original. We speculate that ResNet-18 is capable of extracting deeper pixel-level information, which expands the function class that the policy network can learn from. Furthermore, SPO can still maintain a good probability ratio constraint, thereby benefiting from the theoretical lower bound (10). In contrast, it is challenging to train large neural networks with PPO because the probability ratio can not be controlled during training, even employing a smaller $\epsilon = 0.1$.

### 5.3 CONSTRAINING RATIO DEVIATION

We are not the first to attempt to constrain the differences between successive policies during training. For instance, TRPO (Schulman et al., 2015a) explicitly limits the KL divergence to ensure trust region constraints, while TR-PPO (Wang et al., 2020) employs KL divergence-based clipping objective. Empirically, there have also been efforts to design adaptive learning rates to prevent aggressive policy updates (Achiam et al., 2017; Heess et al., 2017; Queeney et al., 2021; Rudin et al., 2022).

The most successful adaptive learning rates are based on KL divergence, which has been empirically proven to be highly effective (Rudin et al., 2022). Additionally, TR-PPO (Wang et al., 2020) offers an efficient approach for constraining successive policies. We demonstrate their ratio deviations in the Humanoid-v4 environment, employing a seven-layer policy network. Furthermore, we compare them with PPO utilizing a smaller $\epsilon$, results are shown in Fig. 8.

It can be observed that naively reducing $\epsilon$ does not match the ratio deviation curve of SPO. In contrast, KL divergence-based learning rate effectively constrains the ratio deviations of PPO and improves its perfor-

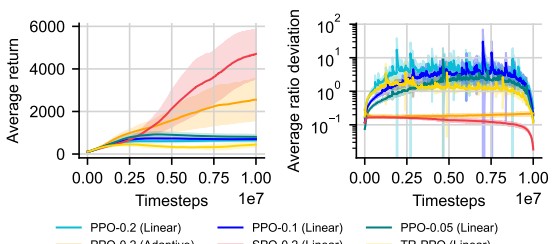

Figure 8: Ratio deviation curves and training performance of SPO and other methods. Where "linear" denotes a linear decay and "adaptive" denotes an adjustment based on the KL divergence.

mance, though it often result in overly conservative policy updates (with the learning rate nearly maintained at the minimum value of 1e-5 during training). Additionally, while TR-PPO can mitigate PPO's ratio deviations to some extent, it does not contribute to performance improvement.

In summary, despite the efforts made by the aforementioned work to constrain the probability ratio between successive policies, the constraints on the probability ratio during training are (i) either limited, resulting in insignificant performance improvements, or (ii) they significantly constrain the probability ratio but lead to overly conservative policy updates. In contrast, SPO stands out with its simple implementation and superior performance, positioning itself as an efficient and promising alternative for model-free policy optimization algorithms.

## 6 CONCLUSION

In this paper, we introduced *Simple Policy Optimization* (SPO), a novel unconstrained first-order algorithm that effectively combines the strengths of Trust Region Policy Optimization (TRPO) and Proximal Policy Optimization (PPO). By incorporating TV divergence into the surrogate objective, SPO maintains optimization within the trust region, benefiting from TRPO's theoretical guarantees while preserving the efficiency of PPO. Our experimental results demonstrate that SPO can achieve competitive performance across various benchmarks with a simple implementation. Moreover, SPO can simplify the training of deep policy networks, addressing a key challenge faced by existing algorithms. These findings indicate that SPO is a promising approach for advancing model-free reinforcement learning. In future work, SPO holds potential for impactful applications in areas such as autonomous driving, robotic control, game AI, and financial modeling. With further research and refinement, we believe, SPO could drive innovation and breakthroughs across these fields.

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

## A    RELATED WORK

Previous studies have explored policy improvement within the trust region. For instance, Queeney et al. (2021) developed an off-policy sample reuse method that combines the stability of on-policy approaches with the sample efficiency of off-policy methods, implementing a mechanism similar to the clipping mechanism in PPO. Achiam et al. (2017) proposed constrained policy optimization (CPO), which restricts policy improvement to a safe region, based on new theoretical results. There has also been work focused on developing novel clipping mechanisms to prevent aggressive policy updates (Wang et al., 2020; Cheng et al., 2021). In terms of implementation, some studies have designed adaptive learning rates based on TV divergence or KL divergence (Heess et al., 2017; Queeney et al., 2021; Rudin et al., 2022), which have been shown to effectively enhance the stability of clipping objectives. Additionally, some work has explored non-parametric methods for policy optimization (Vuong et al., 2018; Song et al., 2019). Empirically, code-level optimizations have proven to be effective in improving the performance of PPO and the implementation details of PPO have been widely adopted (Huang et al., 2022a;b; Dhariwal et al., 2017).

## B    IMPLEMENTATION DETAILS

Our code implementation is primarily based on the high-quality open-source reinforcement learning libraries, CleanRL (Huang et al., 2022b), with all code runs on an NVIDIA GeForce RTX 3090. For details on code-level optimizations, please refer to the mujoco and atari folders in the supplementary materials. We emphasize that no further code-level tuning is applied to SPO, except for modifications to the policy loss. Anyone can run our code to reproduce our results.

## C    COMPARING ALGORITHMS

Table 1: Average return of the entire training process across ten separate runs with different random seeds in MuJoCo benchmark, using policy networks with *three* layers.

| Environment | PPO-Clip | PPO-Penalty | SPO | SPU | PPO-RB | TR-PPO | TR-PPO-RB | RPO |
|---|---|---|---|---|---|---|---|---|
| Ant-v4 | 4234.7 | 1684.6 | 3779.7 | 1985.02 | 767.0 | 3795.3 | 2719.4 | **4248.0** |
| HalfCheetah-v4 | 2964.6 | **3287.9** | 2515.9 | 2537.9 | 1262.7 | 2664.8 | 2846.1 | 3095.2 |
| Hopper-v4 | 1306.9 | 1221.0 | **1507.5** | 1381.8 | 549.2 | 1347.2 | 1120.7 | 1311.3 |
| Humanoid-v4 | 778.5 | 1089.2 | **1808.3** | 1514.8 | 852.9 | 838.5 | 869.2 | 884.3 |
| HumanoidStandup-v4 | 139434.6 | 120696.0 | **150395.0** | 143019.9 | 124482.4 | 141321.6 | 142652.0 | 143434.4 |
| Walker2d-v4 | **3025.1** | 2206.6 | 2738.2 | 2102.7 | 644.3 | 2901.6 | 2279.2 | 2933.1 |
| Total average return | 25290.7 | 21697.6 | **27124.1** | 25423.7 | 21426.4 | 25478.2 | 25414.4 | 25984.4 |

## D    HYPERPARAMETER SETTINGS

Table 2: Detailed hyperparameters in Atari 2600 and MuJoCo.

| Hyperparameters | Atari 2600 (Bellemare et al., 2013) | MuJoCo (Todorov et al., 2012) |
|---|---|---|
| Number of actors | 8 | 8 |
| Horizon | 128 | 256 |
| Learning rate | $2.5 \times 10^{-4}$ | $3 \times 10^{-4}$ |
| Learning rate decay | Linear | Linear |
| Optimizer | Adam | Adam |
| Total steps | $1 \times 10^{7}$ | $1 \times 10^{7}$ |
| Batch size | 1024 | 2048 |
| Update epochs | 4 | 10 |
| Mini-batches | 4 | 32, 4 |
| Mini-batch size | 256 | 64, 512 |
| GAE parameter $\lambda$ | 0.95 | 0.95 |
| Discount factor $\gamma$ | 0.99 | 0.99 |
| Value loss coefficient $c_1$ | 0.5 | 0.5 |
| Entropy loss coefficient $c_2$ | 0.01 | 0.0 |
| Probability ratio parameter $\epsilon$ | 0.2 | 0.2 |

# E SENSITIVITY ANALYSIS

We conducted a sensitivity analysis on the hyperparameter $\epsilon$ of SPO, setting it to 0.1, 0.2, 0.3, 0.4 and 0.5, respectively, the results are shown in Tab. 3. We can see that the best performance of SPO is usually achieved when $\epsilon$ is set to 0.1 and 0.2, while SPO-0.5 usually performs the worst, as it is too loose for the probability ratio constraint in the performance difference lower bound (10).

Similar to Tab. 5, we also present the largest value of average ratio deviation during training, which can be seen from Tab. 4. It is clear that the PPO's ratio deviation is not under control, as its maximum ratio deviation in most environments exceeds a staggering 1000, which is much larger than the expected value of $\epsilon = 0.2$. That's why in Tab. 3, even though SPO-0.5 performs the worst, it is still better than PPO in all environments. We visualize the ratio deviation curves in the Humanoid-v4 environment, which can be seen in Fig. 9. It is clear that the ratio deviation of SPO can be effectively controlled by the hyperparameter $\epsilon$.

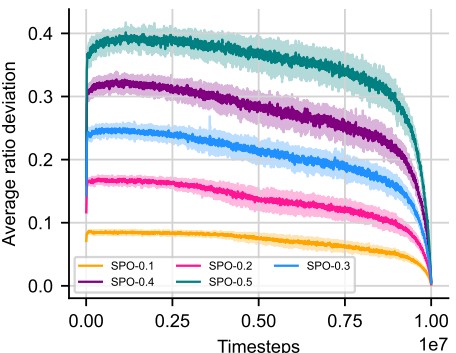

Figure 9: Training curves of ratio deviation of SPO with different $\epsilon$ in Humanoid-v4 environment. The mean and standard deviation are shown across five random seeds.

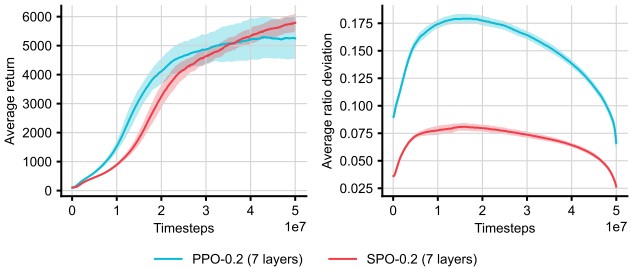

Figure 10: Training performance of PPO and SPO with super large batch size in Humanoid-v4, where batch size and mini-batch size are set to **65536** and **32768**, while all other hyperparameters remain consistent. The mean and standard deviation are shown across five random seeds.

Table 3: Average return of SPO with different $\epsilon$ of the entire training process across five separate runs with different random seeds in MuJoCo benchmark, using policy networks with *seven* layers.

| Environment | SPO-0.1 | SPO-0.2 | SPO-0.3 | SPO-0.4 | SPO-0.5 | PPO-0.2 |
|---|---|---|---|---|---|---|
| Ant-v4 | 3013.7 | **3222.4** | 3176.4 | 2925.7 | 2701.7 | 928.0 |
| HalfCheetah-v4 | 1903.8 | **3764.6** | 2842.6 | 3358.9 | 2522.8 | 1950.8 |
| Hopper-v4 | **1649.3** | 1461.2 | 1237.7 | 1149.2 | 1351.4 | 1107.1 |
| Humanoid-v4 | 2202.1 | **2637.0** | 2457.0 | 2510.6 | 1531.9 | 595.1 |
| HumanoidStandup-v4 | 141551.8 | **154103.6** | 145723.4 | 148864.2 | 139806.3 | 96899.1 |
| Walker2d-v4 | **2776.9** | 2572.3 | 2555.8 | 2178.7 | 1845.4 | 1098.5 |

Table 4: The largest value of average ratio deviation of SPO with different $\epsilon$ during the entire training process across five separate runs with different random seeds in MuJoCo benchmark, using policy networks with *seven* layers. The maximum value in each column is bolded.

| Environment | SPO-0.1 | SPO-0.2 | SPO-0.3 | SPO-0.4 | SPO-0.5 | PPO-0.2 |
|---|---|---|---|---|---|---|
| Ant-v4 | **0.126** | 0.190 | 0.328 | 0.519 | 0.597 | 548.060 |
| HalfCheetah-v4 | 0.121 | 0.188 | 0.381 | **0.528** | 0.591 | 1675.340 |
| Hopper-v4 | 0.094 | **0.194** | **0.548** | 0.334 | 0.555 | 113.178 |
| Humanoid-v4 | 0.095 | 0.191 | 0.367 | 0.378 | 0.452 | 2411.845 |
| HumanoidStandup-v4 | 0.093 | 0.187 | 0.281 | 0.382 | 0.505 | **4018.718** |
| Walker2d-v4 | 0.096 | 0.157 | 0.272 | 0.393 | **0.604** | 998.101 |

# F PROOF OF THEOREM 2.1

According to the definition of unnormalized discounted visitation frequencies, we have

$$
\begin{aligned}
&\mathbb{E}_{s\sim\rho_{\tilde{\pi}}(\cdot),a\sim\tilde{\pi}(\cdot|s)}\left[A_\pi(s,a)\right]\\
&=\sum_s\sum_{t=0}^\infty\gamma^t\mathbb{P}(s_t=s|\tilde{\pi})\sum_a\tilde{\pi}(a|s)\cdot A_\pi(s,a)\\
&=\sum_{t=0}^\infty\sum_s\mathbb{P}(s_t=s|\tilde{\pi})\sum_a\tilde{\pi}(a|s)\cdot\gamma^t A_\pi(s,a)\\
&=\mathbb{E}_{\tau\sim\tilde{\pi}}\left[\sum_{t=0}^\infty\gamma^t A_\pi(s_t,a_t)\right]\\
&=\mathbb{E}_{\tau\sim\tilde{\pi}}\left\{\sum_{t=0}^\infty\gamma^t\left[r(s_t,a_t)+\gamma V_\pi(s_{t+1})-V_\pi(s_t)\right]\right\}\\
&=\mathbb{E}_{\tau\sim\tilde{\pi}}\left\{\sum_{t=0}^\infty\gamma^t r(s_t,a_t)+\sum_{t=0}^\infty\gamma^{t+1}V_\pi(s_{t+1})-\sum_{t=0}^\infty\gamma^t V_\pi(s_t)\right\}\\
&=\mathbb{E}_{\tau\sim\tilde{\pi}}\left\{\sum_{t=0}^\infty\gamma^t r(s_t,a_t)-V_\pi(s_0)\right\}\\
&=\eta(\tilde{\pi})-\eta(\pi),
\end{aligned}
\tag{22}
$$

concluding the proof.

# G TV DIVERGENCE AND PROBABILITY RATIO

**Assumption G.1.** Given the current policy $\tilde{\pi}$ and the old policy $\pi$, we assume that the support of $\tilde{\pi}$ is contained in the support of $\pi$.

According to the definition of TV divergence and Assumption G.1, we have

$$
D_{\mathrm{TV}}^{\max}(\pi,\tilde{\pi})^2=\left[\max_s\frac{1}{2}\sum_{a\in\mathcal{A}}|\tilde{\pi}(a|s)-\pi(a|s)|\right]^2=\frac{1}{4}\left[\max_s\sum_{a\in\mathcal{A}}\pi(a|s)\left|\frac{\tilde{\pi}(a|s)}{\pi(a|s)}-1\right|\right]^2, \tag{23}
$$

then, following the definition of expectation,

$$
D_{\mathrm{TV}}^{\max}(\pi,\tilde{\pi})^2=\frac{1}{4}\max_s\left[\mathbb{E}_{a\sim\pi(\cdot|s)}\left|\frac{\tilde{\pi}(a|s)}{\pi(a|s)}-1\right|\right]^2. \tag{24}
$$

# H    TRAINING CURVES ON ATARI 2600 ENVIRONMENTS

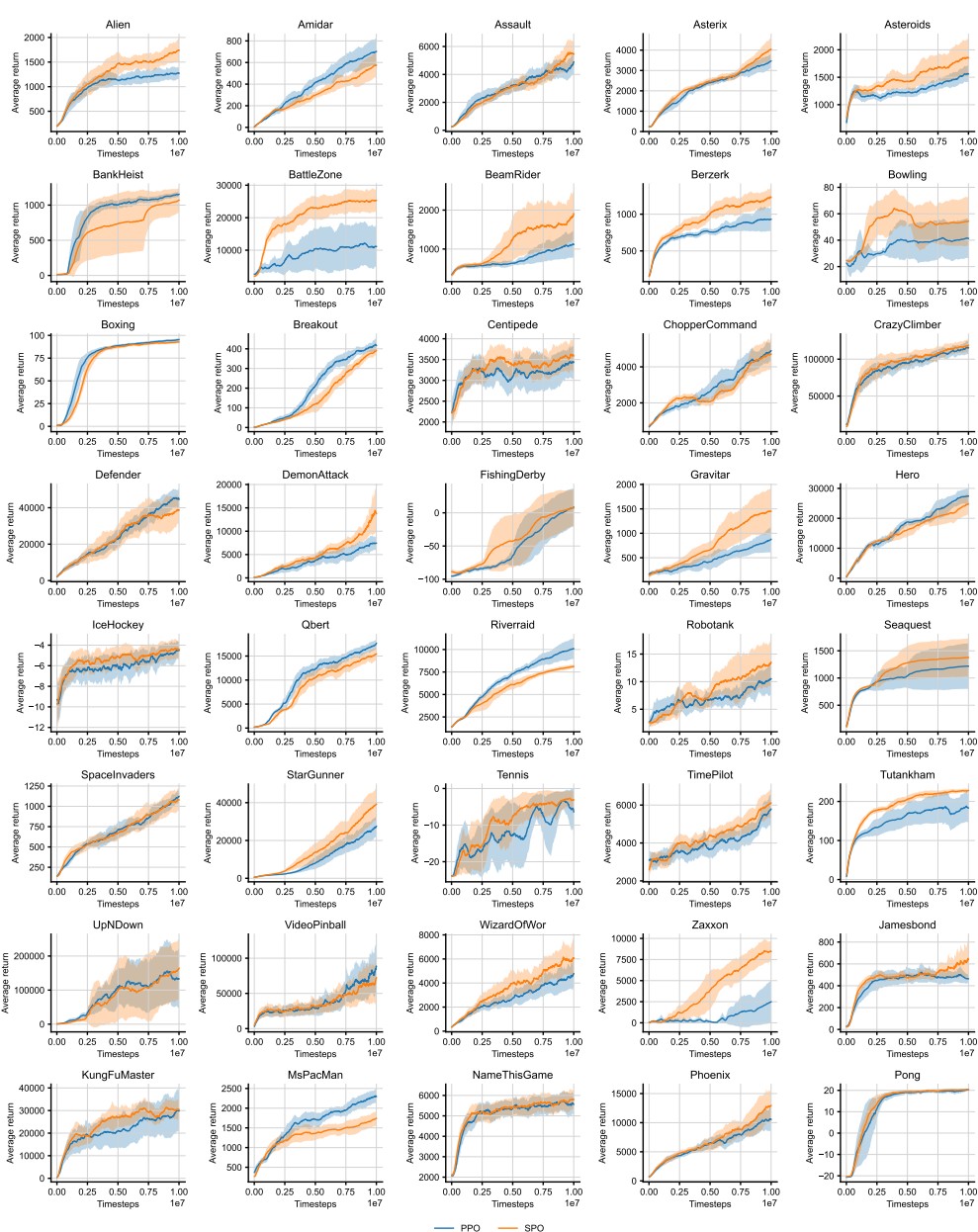

Figure 11: Training performance of PPO and SPO in Atari 2600 benchmark. The mean and standard deviation are shown across three random seeds.

# I MORE EXPERIMENTAL RESULTS

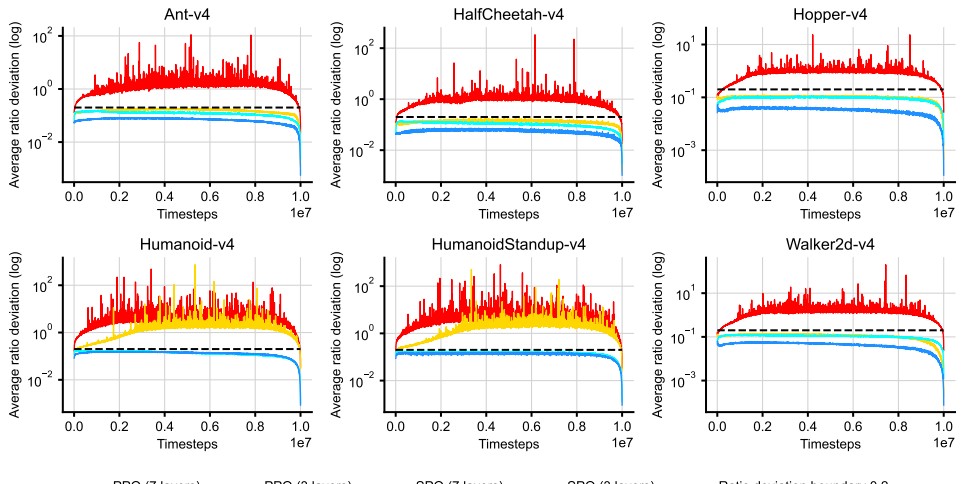

Figure 12: Ratio deviation curves of PPO and SPO with different policy network layers in MuJoCo benchmark. The mean is shown across five random seeds.

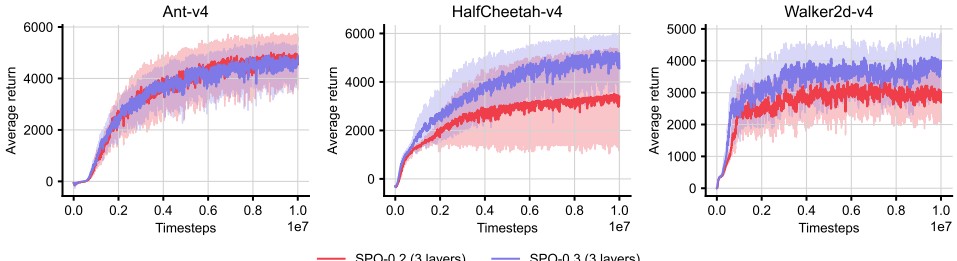

Figure 13: Training performance of SPO with different $\epsilon$ in MuJoCo benchmark. The mean and standard deviation are shown across five random seeds.

Table 5: Average return of PPO and SPO in the last 10% training steps across five separate runs with different random seeds, with their maximum ratio deviation of the entire training process.

| Environment | Index | 3 layers | | 7 layers | |
|---|---|---|---|---|---|
| | | PPO | SPO | PPO | SPO |
| Ant-v4 | Average return (↑) | **5323.2** | 4911.3 | 1002.8 | **4672.5** |
| | Ratio deviation (↓) | 0.229 | **0.101** | 548.060 | **0.190** |
| HalfCheetah-v4 | Average return (↑) | **4550.2** | 3602.4 | 2242.3 | **5307.3** |
| | Ratio deviation (↓) | 0.225 | **0.086** | 1675.340 | **0.188** |
| Hopper-v4 | Average return (↑) | 1119.4 | **1480.3** | 975.9 | **1507.6** |
| | Ratio deviation (↓) | 0.164 | **0.067** | 113.178 | **0.194** |
| Humanoid-v4 | Average return (↑) | 795.1 | **2870.0** | 614.1 | **4769.9** |
| | Ratio deviation (↓) | 3689.957 | **0.179** | 2411.845 | **0.191** |
| HumanoidStandup-v4 | Average return (↑) | 143908.8 | **152378.7** | 92849.7 | **176928.9** |
| | Ratio deviation (↓) | 2547.499 | **0.182** | 4018.718 | **0.187** |
| Walker2d-v4 | Average return (↑) | **3352.3** | 2870.2 | 1110.9 | **3008.1** |
| | Ratio deviation (↓) | 0.170 | **0.070** | 998.101 | **0.157** |