# OpenReview forum: "Simple Policy Optimization"
_ICLR.cc/2025/Conference — ICLR 2025 Conference Withdrawn Submission_

### Official Review · Reviewer_Dgb4 · 2024-10-28

**Soundness:** 4
**Presentation:** 4
**Contribution:** 3
**Rating:** 8
**Confidence:** 3

**Summary:**

The paper introduces a new RL algorithm, Simple Policy Optimisation (SPO), designed to address the challenges present in existing policy gradient methods, particularly PPO and TRPO. The primary goal is to develop a surrogate objective that constrains the policy ratio more effectively than PPO without the computational burden of TRPO's second-order optimisations. SPO achieves this by using TV divergence, combining TRPO's theoretical robustness with PPO's simplicity, and is demonstrated to improve performance across Atari and MuJoCo benchmarks.

**Strengths:**

1. **Clarity and Theory**: The paper is clearly written and provides strong theoretical grounding. New theorems and proofs substantiate the methodology, ensuring that the approach is technically sound.
2. **Experimental Rigor**: Extensive experiments on various benchmarks, including a range of baselines, demonstrate SPO's performance relative to other algorithms. The inclusion of pseudo-code and naming the used open-source repositories contributes to the reproducibility of the work. Additionally, hyperparameters are provided.
3. **Algorithm Simplicity and Impact**: SPO is a straightforward modification of PPO by changing the surrogate objective slightly, achieving meaningful performance improvements, particularly in terms of ratio constraint handling. The simplicity of SPO, combined with its impact, makes it valuable for the community.
4. **Comparative Baselines**: The paper employs a wide array of baselines (for the Mujoco tasks at least), giving context to the results and situating SPO’s benefits accurately. Understandably, Atari is very computationally expensive so it is hard to baseline everything as done in Mujoco so I am not too fazed although I would have liked 1 additional algorithm.

**Weaknesses:**

1. **Lack of Confidence Intervals**: The experimental results lack confidence intervals, which limits the ability to assess the performance variance across runs.
2. **Absence of Statistical Tests**: No statistical testing (such as that from the [rliable](https://github.com/google-research/rliable) package) was performed, making it difficult to confirm the statistical significance of the improvements over PPO and other methods. This draws concern as to whether the improvements are statistically significant.
3. **Potential Baseline Tuning Issues**: PPO and similar RL algorithms are known to be highly sensitive to hyperparameters like batch size, mini-batch size, and the number of rollout steps. The paper does not discuss any dedicated hyperparameter tuning for the baselines, raising concerns about potential underperformance. It was noted that hyperparameters were used from the original papers but this is not always a fair approach especially considering certain types of parameters are not always listed such as number of rollout workers.
4. **Code-level Optimisations**: As noted in the paper, code level optimisations can significantly impact the performance of certain algorithms, but the paper does not specify what implementation tricks were used in the baselines and SPO itself.

**Questions:**

**Questions for the Authors**
1. Could you provide confidence intervals and other statistical measures for the experimental results to help clarify performance variance and statistical significance?
2. Were any efforts made to tune the baseline hyperparameters specifically? If so, could you detail the tuning process? Additionally, how much effort was made tuning SPO?

Ultimately, I liked the paper, think it is a simple yet efficient algorithm and I believe it would be useful and warrants acceptance **if just slightly more validated results are presented.** This should be as simple as presenting the mean, median and interquartile mean of the results as well as using stratified bootstrapping to produce confidence intervals i.e. no further training runs should be done. Due to this, I am giving it a 5 as I do not believe the results can be presented as it currently stands. I look forward to raising the score.

---

> ### Author Response · Authors · 2024-11-19
>
> Dear Reviewer Dgb4,
>
> We are truly grateful for your thoughtful review of our paper and for the encouraging support you've shown towards our work! In the following, we will provide a comprehensive response to your comments.
>
> To address your concerns, we have submitted **all the code for PPO and SPO** used in this paper as supplementary material. Additionally, a **new version of the paper** has been uploaded, with the main revisions **highlighted in red** for your convenience in further review.
>
> # I. Results Lack of Statistical Testing
>
> > **Lack of Confidence Intervals and Statistical Tests**
>
> > **Could you provide confidence intervals and other statistical measures for the experimental results to help clarify performance variance and statistical significance?**
>
> We sincerely appreciate your thoughtful consideration. Based on your suggestion, we have referred to [1] and plotted the **aggregate metrics** on Mujoco-v4 with 95% confidence intervals based on data from **6 environments (Figure 5) using PPO-normalized score**. We found that SPO achieved competitive performance across various statistical measures, which enhances the paper's persuasiveness.
>
> # II. Potential Baseline Tuning Issues
>
> > **PPO and similar RL algorithms are known to be highly sensitive to hyperparameters like batch size, mini-batch size, and the number of rollout steps. The paper does not discuss any dedicated hyperparameter tuning for the baselines, raising concerns about potential underperformance.**
>
> Your feedback is extremely beneficial to us. To address your concerns, we referred to the open-source standard reinforcement learning library, CleanRL [2], and attempted to **align all hyperparameters**. We found that the only difference was in the **mini-batch size** setting, where our paper defaults to 512. Therefore, we focused on comparing the performance differences between SPO and PPO under different **network depths and mini-batch size** settings, and we supplemented the results with statistical tests, as shown in **Figure 7**. We found, as you mentioned, that **PPO is quite sensitive to hyperparameters and network depth**. In contrast, SPO showed significant improvements across various statistical measures and demonstrated relative robustness to mini-batch size. We hope this addresses your concerns!
>
> # III. Code-Level Optimizations
>
> > **As noted in the paper, code level optimisations can significantly impact the performance of certain algorithms, but the paper does not specify what implementation tricks were used in the baselines and SPO itself.**
>
> > **Were any efforts made to tune the baseline hyperparameters specifically? If so, could you detail the tuning process? Additionally, how much effort was made tuning SPO?**
>
> Thank you for your question! We want to emphasize that for all the comparative experiments involving SPO and PPO in the paper, the **only difference is the policy loss**. You can also refer to the code implementation for Mujoco and Atari in the **supplementary material**, where the **only difference** is in the `compute_policy_loss` function within the `trainer.py` file. No further code-level tuning is applied to SPO, highlighting its simplicity and efficiency. For code-level details, our implementation is primarily based on [2] and [3], where these details are thoroughly explained. **The code details for SPO are completely consistent with those for PPO**.
>
> # IV. Additional Supplementary Experiments
>
> Some additional interesting experiments have been added :), as shown in **Figures 8 and 9**. We found that naively reducing PPO's $\epsilon$ does not prevent excessive ratio deviation, further indicating that SPO's effective constraint on the probability ratio is non-trivial.
>
>
> # **References**
>
> [1] R Agarwal et al. Deep reinforcement learning at the edge of the statistical precipice. https://arxiv.org/abs/2108.13264
>
> [2] S Huang et al. Cleanrl: High-quality single-file implementations of deep reinforcement learning algorithms. https://github.com/vwxyzjn/cleanrl
>
> [3] S Huang et al. The 37 implementation details of proximal policy optimization. https://iclr-blog-track.github.io/2022/03/25/ppo-implementation-details/

---

> > ### Comment · Reviewer_Dgb4 · 2024-11-19
> > **Response**
> >
> > Thank you for addressing my concerns. I now feel comfortable raising my score and believe the paper warrants acceptance. Last small thing I would mention is simply that I think the aggregate metrics plots can be made more visually appealing through slightly manipulating the margins etc. Its a little hard to see the performance at a glance. However this is just a minor comment.

---

> > > ### Author Response · Authors · 2024-11-19
> > >
> > > Dear Reviewer Dgb4,
> > >
> > > Thank you once again for your support of our work. We will make efforts to adjust the image margins to ensure they are visually appealing.

---

### Official Review · Reviewer_ESWe · 2024-10-29

**Soundness:** 3
**Presentation:** 4
**Contribution:** 3
**Rating:** 8
**Confidence:** 4

**Summary:**

This paper introduces an actor-critic algorithm, Simple Policy Optimization (SPO), which achieves a more tight bound on the policy _probability ratio_ by modifying the loss function of Proximal Policy Optimization (PPO) algorithm. The authors demonstrate both theoretically and empirically that SPO imposes a stricter upper bound on the _ratio deviation._ Notably, SPO outperforms PPO across various tasks, especially when applied to policies with deeper network architectures. Although several papers have already pointed out issues with the heuristic clipping technique in PPO and proposed alternatives, I find the authors' approach offers a novel perspective, leading me to lean towards acceptance.

**Strengths:**

* This paper is well-written and clearly structured.
* The theoretical foundation is straightforward (in a positive way), and the implementation is simple, which could enhance reproducibility.
* SPO demonstrates superior performance over PPO, especially when using deeper networks (e.g., 7-layer MLP and ResNet).

**Weaknesses:**

* SPO tends to make the _ratio deviation_ overly small in cases with fewer layers (see Table 2). Could the authors address whether the tighter ratio deviation in SPO may be limiting performance in some simpler tasks, and if so, how this trade-off might be managed?

* More importantly, without a comparison to existing methods like TR-PPO (as mentioned by the authors / Line 90), it is hard to assess how effective SPO is relative to other approaches that aim to address the same clipping issue in PPO. Could the authors include a comparison of SPO with TR-PPO and other recent methods addressing PPO's clipping issues, particularly focusing on performance and ratio deviation metrics across different network depths?

**Questions:**

**[Q1]**
In Table 2, SPO does not outperform PPO with a 3-layer network. Could you clarify this? In tasks where SPO underperforms (e.g., Ant, HalfCheetah, Walker2d), the ratio deviation appears to be constrained overly tightly.

**[Q2]**
To me, the transition from Equation (10) to Equation (14) feels somewhat abrupt. Could you elaborate further on this derivation? I understand the motivation behind Equation (14), presumably to demonstrate the bounded nature of the probability ratio as indicated by Theorem 4.3. However, if you could clarify how the $C$ and the max operator in Equation (10) relate to Equation (14), it would provide stronger theoretical support.

**[Q3]**
Is SPO also more applicable to large-scale scenarios and does it still show superiority in terms of ratio deviation? I’m curious about its performance difference from PPO in cases with extensive parallelism, such as environments with many actors like Isaac Gym.

**[Q4]**
When using a 3-layer network, what is the performance difference between SPO-0.3 and SPO-0.2? If SPO-0.2 constrains the ratio deviation overly tightly, causing it to underperform compared to PPO, SPO-0.3 might perform better due to the slightly looser upper bound on ratio deviation. If this is the case, it would provide stronger theoretical support for the approach.

---

> ### Author Response · Authors · 2024-11-21
>
> Dear Reviewer ESWe,
>
> We are truly grateful for your thoughtful review of our paper and for the encouraging support you've shown towards our work! In the following, we will provide a comprehensive response to your comments.
>
> Additionally, we would like to gently remind you that **all code implementations of SPO and PPO** used in our work have been uploaded to the supplementary materials. The revised version of the paper has also been uploaded, with the main revisions **highlighted in red** for your further review.
>
> # I. The Constraint on the Ratio Deviation
>
> >**SPO tends to make the ratio deviation overly small in cases with fewer layers (see Table 2). Could the authors address whether the tighter ratio deviation in SPO may be limiting performance in some simpler tasks, and if so, how this trade-off might be managed?**
>
> >**SPO does not outperform PPO with a 3-layer network. Could you clarify this? In tasks where SPO underperforms (e.g., Ant, HalfCheetah, Walker2d), the ratio deviation appears to be constrained overly tightly.**
>
> >**When using a 3-layer network, what is the performance difference between SPO-0.3 and SPO-0.2? If SPO-0.2 constrains the ratio deviation overly tightly, causing it to underperform compared to PPO, SPO-0.3 might perform better due to the slightly looser upper bound on ratio deviation. If this is the case, it would provide stronger theoretical support for the approach.**
>
> - These issues directly hit the point! We found that the policy network with fewer parameters exhibits **relatively conservative ratio deviation** during training (see **Figure 13**). This indeeds limit the performance of SPO on some simpler tasks (as shown in **Figure 6**). The 3-layer MLP used in the experiments, with hidden layers of [64, 64] (based on the default settings from CleanRL [1] and most papers), has very **limited expressive power**. Our idea is that simply increasing the network parameters can effectively address this issue (Figure 8 somewhat proves this point).
>
> - Then, based on your suggestion, we conducted experiments with **a larger $\epsilon$** on Ant-v4, HalfCheetah-v4, and Walker2d-v4 (see **Figure 14** on page 18), and we found that the results **align well with your expectations**. This suggests that the setting of $\epsilon$ indeed involves a trade-off: smaller epsilon leads to more stable performance improvement (based on the lower bound of policy improvement), but it may also make the policy updates more conservative. Our current response to this is to recommend using a **relatively small $\epsilon$** to ensure robust policy improvement, while selecting neural networks with as many parameters as possible to provide **sufficient expressive power**.
>
> # II. Comparison to Existing Methods
>
> >**Could the authors include a comparison of SPO with TR-PPO and other recent methods addressing PPO's clipping issues, particularly focusing on performance and ratio deviation metrics across different network depths?**
>
> Certainly! Following your suggestion, we added an **additional section** and compared PPO with a smaller $\epsilon$ under the same settings, as well as adaptive learning rates [2] and the TR-PPO [3] you mentioned. These methods also help to some extent in limiting aggressive policy updates, and the results can be found in **Figure 9** on page 10. We found that naively using a smaller $\epsilon$ does not alleviate the issue of excessive ratio deviation in PPO. Furthermore, while adaptive learning rates do limit the ratio deviation, we observed that they lead to **overly conservative policy updates** (we noticed that throughout the experiment, the learning rate almost remained at 1e-5).

---

> ### Author Response · Authors · 2024-11-21
>
> # III. Theoretical Derivation
>
> >**To me, the transition from Equation (10) to Equation (14) feels somewhat abrupt. Could you elaborate further on this derivation?**
>
> Sure! Let us first write down the lower bound of performance improvement $$\eta(\tilde{\pi})-\eta(\pi)\geq E_{s\sim{\rho_{\pi}(\cdot)},a\sim{\pi(\cdot|s)}}\left[\frac{\tilde{\pi}(a|s)}{\pi(a|s)}\cdot A_{\pi}(s,a)\right]-\frac{C}{4}\max_{s}\left[{E}_{a\sim\pi(\cdot|s)}\left|\frac{\tilde{\pi}(a|s)}{\pi(a|s)}-1\right|\right]^2=\mathcal{L}$$
> where $C=\frac{4\epsilon\gamma}{(1-\gamma)^2}$ is unknown for us. Therefore, it is impossible for us to **directly optimize** the lower bound $\mathcal{L}$. The commonly used approach is to attempt to constrain the **latter term** while maximizing the **former term** (the surrogate objective).
>
> Considering that the **maximization operation** of the latter term requires traversing **all states $s\in\mathcal{S}$**, this is also not feasible. Therefore, we can only empirically impose the ratio constraint on each current data point $(s_t,a_t)\sim\pi_{\theta_{\mathrm{old}}}$, which is
> $$\left|\frac{\pi_{\theta}(a_t|s_t)}{\pi_{\theta_{\mathrm{old}}}(a_t|s_t)}-1\right|\leq\epsilon,\enspace\forall (s_t,a_t)\sim\pi_{\theta_{\mathrm{old}}}$$
> and this leads us to derive the **optimization problem (15)** in our current paper, which is
> $$\max_{\theta}\enspace E_{(s_t,a_t)\sim\pi_{\theta_{\mathrm{old}}}}\left[\frac{\pi_{\theta}(a_t|s_t)}{\pi_{\theta_{\mathrm{old}}}(a_t|s_t)}\cdot A(s_t,a_t)\right]$$
> $$\mathrm{s.t.}\enspace\left|\frac{\pi_{\theta}(a_t|s_t)}{\pi_{\theta_{\mathrm{old}}}(a_t|s_t)}-1\right|\leq\epsilon,\enspace\forall (s_t,a_t)\sim\pi_{\theta_{\mathrm{old}}}$$
>
> Simplifying the notation, let the probability ratio be denoted as $r$, and the advantage function as $A$. Then, we aim to obtain an objective function $f(r,A,\epsilon)$ (where $r$ is the **optimization variable**) that maximizes $rA$ while constraining $|r-1|\leq\epsilon$. Additionally, we want this function to be **convex**, so why not consider a **quadratic polynomial** in $r$? For simplicity, let's assume $$f(r,A,\epsilon)=rA-k\cdot (r-1)^2$$
> This is a quadratic function of $r$, with an unknown coefficient $k$. We also want the **optimal solution to be at the boundary of our constraint** $r=1+\mathrm{sign}(A)\cdot\epsilon$. By solving this simple differential equation, we obtain our objective $$f(r,A,\epsilon)=rA-\frac{|A|}{2\epsilon}\cdot (r-1)^2$$
> we presented this very intuitive approach to the derivation.
>
> # IV. SPO in Large-Scale Scenarios
>
> >**Is SPO also more applicable to large-scale scenarios and does it still show superiority in terms of ratio deviation? I’m curious about its performance difference from PPO in cases with extensive parallelism.**
>
> Thank you for your suggestion. We have included an experimental comparison of PPO and SPO with a **large mini-batch size of 32,768** (see **Figure 11** on page 15). We were surprised to find that with a large amount of data, the ratio deviation of PPO is **well constrained** (indicating that large-scale training data seems to mitigate excessive ratio deviation in PPO). In contrast, SPO generally has **smaller variance and robust performance**, and often matches or even surpasses PPO's performance in the later stages of training.
>
> Additionally, for **training with very large batch sizes**, we recommend using a **larger** $\epsilon$ for SPO to match the early training advantages of PPO. We are also eager to further explore the use of **adaptive $\epsilon$** in SPO during training to combine PPO's sample efficiency in the early stages with SPO's robust performance in the later stages.
>
> # Thank you again for your valuable suggestions and for your support of our work
>
> # References
>
> [1] S Huang et al. Cleanrl: High-quality single-file implementations of deep reinforcement learning algorithms. https://github.com/vwxyzjn/cleanrl
>
> [2] N Rudin et al. Learning to walk in minutes using massively parallel deep reinforcement learning. https://proceedings.mlr.press/v164/rudin22a.html
>
> [3] Y Wang et al. Truly proximal policy optimization. https://proceedings.mlr.press/v115/wang20b.html

---

> > ### Comment · Reviewer_ESWe · 2024-11-21
> > **Thank you**
> >
> > Thank you very much for your detailed response and for taking the time to conduct the additional experiments. SPO seems to be very well-suited for visual RL since large networks are required to effectively extract features from high-dimensional images.

---

> > > ### Author Response · Authors · 2024-11-21
> > >
> > > Dear Reviewer ESWe,
> > >
> > > We are glad to address your concerns. Large-scale visual reinforcement learning sounds like a great suggestion (a small reminder: batch normalization can be harmful to RL)!

---

### Official Review · Reviewer_HKZk · 2024-10-31

**Soundness:** 2
**Presentation:** 2
**Contribution:** 2
**Rating:** 3
**Confidence:** 5

**Summary:**

The paper introduces an on-policy RL algorithm called Simple Policy Optimization (SPO). SPO is proposed as an improvement to PPO that can more effectively constrain the probability ratio between the next and current policy. The objective of SPO is designed to update the probability ratio to the boundary of $[1-\epsilon, 1+\epsilon]$ in the direction that improves the surrogate objective. Experiments on Atari and MuJoCo benchmarks in Gymnasium demonstrate that SPO controls the probability ratio and achieves performance that generalizes across different network architectures, compared to PPO which is more sensitive to implementation choices.

**Strengths:**

- **[S1] Simple implementation:** The implementation of SPO requires minimal changes to PPO and considers a simple unconstrained optimization problem.
- **[S2] Robust performance across architectures:** Experiments show that SPO generalizes well across different implementation choices such as network size, compared to PPO which is much more sensitive to these choices. Given the widespread use of PPO, improvements that provide robustness to hyperparameters / architectures are helpful for applying RL training across different applications.

**Weaknesses:**

**[W1] Limited novelty compared to prior works, with several missing citations to relevant prior works**
- The results presented in the paper have limited theoretical novelty. Section 3 uses results that have already been shown in [1] to rewrite the TV distance, and applies Pinsker’s inequality in ways that have been used previously in [2]. The optimization problem in (15) is also not novel, and can be motivated from the policy improvement bounds proposed in [2] that include an expected TV distance penalty term (instead of a max over states). Section 4 formulates an objective that satisfies the set of assumptions put forth by the authors in Definition 4.1.
- Several methods have been proposed to better constrain the probability ratio in PPO, and SPO represents another implementation of this same idea. [1, 3] apply adaptive learning rates based on TV or KL divergence, [4, 5] propose modifications to the clipping function / rollback mechanisms, and [6, 7] consider non-parametric updates that determine target probability ratios. The SPO objective in (14) is designed to move each data point towards the target probability ratio in Definition 4.1, which is similar to what is accomplished by many of these methods.
- Most of the relevant works mentioned in the 2 previous bullet points are never referenced in the paper ([4] is the only work that is cited in the paper / compared against in the experiments). In general, the paper does not adequately frame its contribution with respect to prior work (there is no Related Work section in the paper).

**[W2] Experiments: contributions are overstated and suboptimal PPO implementation is used for comparison**
- At several points throughout the paper, the authors claim that SPO achieves state-of-the-art performance. However, the results shown in the paper are not state-of-the-art. Several off-policy algorithms have been shown to perform much better than SPO (and on-policy algorithms in general) on these benchmarks, and in some cases it is possible to find other implementations of PPO that achieve better performance than what has been reported in this paper (see the results contained in the references below).
- Experiments compare against PPO for a single set of hyperparameters / implementation choices that may be suboptimal based on other results reported in the literature. While I understand that this shows PPO can be more sensitive to implementation details than SPO, it would be much more convincing to show that SPO can perform similar to a well-tuned version of PPO across different tasks / architectures.

**Minor technical issues:**
- I believe that (8) is not correct due to the min that appears in the PPO objective in (6). The cases must also consider the sign of the advantage function (e.g., see (7) in [4]).
- The connection between TV distance and the probability ratio in (9) / Appendix B requires the assumption that the support of $\tilde{\pi}$ is contained in the support of $\pi$. This technical assumption is not mentioned anywhere in the paper.

---

**References:**

[1] Queeney et al. Generalized Proximal Policy Optimization with Sample Reuse. In NeurIPS 2021.

[2] Achiam et al. Constrained Policy Optimization. In ICML 2017.

[3] Rudin et al. Learning to Walk in Minutes Using Massively Parallel Deep RL. In CoRL 2022.

[4] Wang et al. Truly Proximal Policy Optimization. In UAI 2020.

[5] Cheng et al. Authentic Boundary Proximal Policy Optimization. IEEE Transactions on Cybernetics, 2021.

[6] Vuong et al. Supervised Policy Update for Deep RL. In ICLR 2019.

[7] Song et al. V-MPO: On-Policy Maximum a Posteriori Policy Optimization for Discrete and Continuous Control. In ICLR 2020.

**Questions:**

**[Q1] Can you please explain why each assumption in Definition 4.1 is important?** When comparing the objective in (14) to PPO, it seems like the existence of a unique optimal probability ratio in SPO is actually the important feature compared to multiple optimal solutions in PPO.

**[Q2] Could you please explain the purpose of Section 3 and the optimization problem in (15)?** It does not look like (15) is actually being used by SPO. Instead, the main algorithmic contribution is the objective in (14), which is motivated by considering the optimization problem in (17) at every state-action pair. Applying constraints at every state-action pair is more restrictive than the constraint in (15), and also does not account for the dependence across state-action pairs that occurs because $\pi$ must be a probability distribution.

**[Q3] Experimental suggestions:** I understand that these suggestions may not be possible in a limited rebuttal window.
- How does SPO compare to well-tuned versions of PPO across each setting?
- How does SPO compare to PPO using the adaptive learning rate in [3], which is a simple implementation detail used in the literature to prevent probability ratios from becoming very large?
- How does SPO compare to non-parametric policy update methods such as [6, 7]? Similar to SPO, these methods apply updates based on target probability ratios.

---

> ### Author Response · Authors · 2024-11-20
>
> Dear Reviewer HKZk,
>
> We sincerely thank you for your thorough review and valuable feedback on our paper, which will greatly help us improve the quality of our work. Below, we will address your concerns.
>
> Please note that, in order to address your concerns, we have submitted **all the code for PPO and SPO** used in this paper as supplementary material. Additionally, we have uploaded **a new version of the paper**, with the major revisions **highlighted in red** for your convenience in further review.
>
> # I. Limited Novelty and Lack of Related Work
>
> >**Limited novelty compared to prior works, with several missing citations to relevant prior works**
>
> >**the paper does not adequately frame its contribution with respect to prior work (there is no Related Work section in the paper).**
>
> **(i) Lack of related work**. We appreciate the reviewer for carefully reading and pointing out these two issues. We realize that this work indeed lacks proper citations to related work [1-7]. As you suggested, we have included the references you listed and **added a section on related work (Appendix A)** to introduce prior research. Additionally, we have briefly outlined the **implementation details (Appendix B)**, which addresses concerns from most reviewers regarding the potential difficulty of reproducing the results.
>
> **(ii) Limited novelty**, as you mentioned, previous work has established the relationship between TV divergence and the probability ratio [1], and we fully acknowledge this. However, despite this, we also have our **own unique contributions**:
>
> - Firstly, previous work focused on probability ratio constraints based on **clipping function**, whereas we have thoroughly pointed out the potential drawbacks of clipping function (**Figures 2 and 4**) in our paper. For deeper networks, these drawbacks are **highly detrimental to policy improvement**.
>
> - Secondly, we formally prove why using a trust region based on **TV divergence constrain is a better choice** (**Theorem 3.3**). This is due to the fact that the KL divergence constraint is looser, which leads to a smaller solution space, and the conclusion is non-trivial.
>
> - More importantly, our novel objective provides **theoretical guarantees for constraining probability ratio**. The objective of SPO is **convex with respect to the probability ratio**, and the optimal value is exactly attained at the **constraint boundary**. Therefore, SPO offers strong theoretical guarantees for the probability ratio constraint and only requires slight modifications to the PPO objective to achieve significant performance improvements. In contrast, while previous work has also made efforts to constrain the differences between continuous policies, we believe that the SPO objective is quiet **simple and effective**. Furthermore, **SPO achieves competitive performance improvements (Figures 5, 7, and 8), especially for deeper networks**.
>
> We sincerely appreciate Reviewer HKZk's valueble feedback, which has encouraged us to further clarify the novelty of our work. We would also like to note that other reviewers have recognized the unique contributions of our approach:
> >Reviewer **ESWe** remarked: _"Although several papers have already pointed out issues with the heuristic clipping technique in PPO and proposed alternatives, I find the authors' approach offers a novel perspective, leading me to lean towards acceptance."_
>
> >Reviewer **Dgb4** highlighted: _"SPO is a straightforward modification of PPO by changing the surrogate objective slightly, achieving meaningful performance improvements, particularly in terms of ratio constraint handling. The simplicity of SPO, combined with its impact, makes it valuable for the community."_

---

> ### Author Response · Authors · 2024-11-20
>
> # II. Contributions are Overstated and Suboptimal PPO Implementation
>
> >**At several points throughout the paper, the authors claim that SPO achieves state-of-the-art performance.**
>
> >**it would be much more convincing to show that SPO can perform similar to a well-tuned version of PPO across different tasks / architectures.**
>
> - **Contributions are Overstated**. We sincerely appreciate your thoughtful comments, and we recognize the issue of overstating our contributions. As a result, we have replaced all instances of **"state-of-the-art performance"** in the original paper with **"SPO outperforms PPO"** and **"achieves competitive performance"**, on **line 22, line 105, and line 532** in our current paper.
>
> - **PPO Implementation**. To address your concerns, we **aligned the hyperparameters** with those from high-quality open-source reinforcement learning libraries, CleanRL [8]. We found that all hyperparameters for Atari are consistent, enabling PPO to achieve ~400 points in `Breakout-v4`. Additionally, we compared the hyperparameters for MuJoCo and found that the **only difference** was in the choice of mini-batch size. Based on your suggestion, we included a performance comparison for **different network depths and mini-batch sizes (Figure 7)**. The results show that SPO outperforms PPO across various statistical metrics and demonstrates relative robustness across different network depths and hyperparameter settings. **Additionally, all our code has been uploaded to the supplementary materials** for your review. We hope this addresses your concerns.
>
> # III. Minor Technical Issues
>
> >**I believe that (8) is not correct due to the min that appears in the PPO objective in (6). The cases must also consider the sign of the advantage function.**
>
> >**The connection between TV distance and the probability ratio in (9) / Appendix B requires the assumption that the support of $\tilde{\pi}$ is contained in the support of $\pi$.**
>
> - We appreciate your incredibly detailed review! As you mentioned, the gradient of PPO needs to account for the sign of advantage $\hat{A}_{\pi}(s_t,a_t)$. We have made the necessary modifications to **(8)**, though **this does not affect the subsequent derivations**.
>
> - Considering that typical policy networks usually output softmax-normalized probabilities (for discrete action spaces) or a multivariate Gaussian distribution (for continuous action spaces), we have ignored the assumption you mentioned in these two cases. For theoretical rigor, we have added this assumption in **Appendix G** of the current paper. Thank you for your suggestion.
>
> # IV. Some Problems
>
> >**Can you please explain why each assumption in Definition 4.1 is important? When comparing the objective in (14) to PPO, it seems like the existence of a unique optimal probability ratio in SPO is actually the important feature compared to multiple optimal solutions in PPO.**
>
> >**Could you please explain the purpose of Section 3 and the optimization problem in (15)? It does not look like (15) is actually being used by SPO. Instead, the main algorithmic contribution is the objective in (14), which is motivated by considering the optimization problem in (17) at every state-action pair. Applying constraints at every state-action pair is more restrictive than the constraint in (15), and also does not account for the dependence across state-action pairs that occurs because $\pi$ must be a probability distribution.**
>
> Thank you for your questions; they have inspired us to some extent. We will address your concerns below.
>
> - Firstly, each assumption in Definition 4.1 is important because the inclusion of the $rA$ term ensures that the gradient is indeed **directed towards optimizing the surrogate objective, rather than just moving towards the constraint boundary**. Additionally, $A\neq 0$ is necessary to express the optimal solution, as $A=0$ leads to the trivial case where both PPO and SPO objectives are always zero. The $\epsilon$ term is a constraint on the ratio deviation and thus must be positive. Finally, $f$ and $g$ being convex with respect to $r$ ensures that the gradient optimization algorithm will constrain the probability ratio within the boundary $r=1+\mathrm{sign}(A)\cdot\epsilon$.
>
> - Secondly, this is a **really good question**. The optimization objective in **(15)** should impose ratio deviation constraints for **all** sampled $(s_t,a_t)\sim\pi_{\theta_{\mathrm{old}}}$ (we have **corrected this and highlighted it in red**). Constraining the ratio deviation expectation for all current data **is similar to the heuristic approximation in TRPO**, relaxing the maximization operation in the lower bound **(10)** to an average. Therefore, the current objective **(15)** aligns well with the SPO objective **(14)**, as imposing constraints for each current data point $(s_t,a_t)\sim\pi_{\theta_{\mathrm{old}}}$ more closely aligns with the maximization operation in the lower bound **(10)**.

---

> ### Author Response · Authors · 2024-11-20
>
> # V. Experimental Suggestions
>
> >**How does SPO compare to well-tuned versions of PPO across each setting?**
>
> >**How does SPO compare to PPO using the adaptive learning rate in [3], which is a simple implementation detail used in the literature to prevent probability ratios from becoming very large?**
>
> >**How does SPO compare to non-parametric policy update methods such as [6, 7]? Similar to SPO, these methods apply updates based on target probability ratios.**
>
> - Thank you for your question. To address your concerns, we conducted additional experiments. These include a comparison of SPO and PPO with different network depths and mini-batch sizes (**Figure 7**). For adaptive learning rates, we show several methods you mentioned (**Figure 9**) and created a **new section** for this [3, 4]. We found that while adaptive learning rates improved performance, they often led to **overly conservative policy updates**, as we observed that the learning rate magnitude remained around 1e-5 throughout the learning process.
>
> - Finally, regarding the suggestion to compare with non-parametric learning methods, since our work **primarily focuses on parameterized policy improvement methods**, we did not conduct additional comparative experiments. Nevertheless, we **introduced these non-parametric works in the newly added related work section (Appendix A)**.
>
> # Thank you once again for your detailed review, which has greatly improved the quality of our work.
>
> # Reference
>
> [1] J Queeney et al. Generalized Proximal Policy Optimization with Sample Reuse. https://proceedings.neurips.cc/paper_files/paper/2021/hash/63c4b1baf3b4460fa9936b1a20919bec-Abstract.html
>
> [2] J Achiam et al. Constrained policy optimization. https://proceedings.mlr.press/v70/achiam17a
>
> [3] N Rudin et al. Learning to walk in minutes using massively parallel deep reinforcement learning. https://proceedings.mlr.press/v164/rudin22a.html
>
> [4] Y Wang et al. Truly proximal policy optimization. https://proceedings.mlr.press/v115/wang20b.html
>
> [5] Y Cheng et al. Authentic boundary proximal policy optimization. https://ieeexplore.ieee.org/abstract/document/9376693?casa_token=_a37KD3Uk6YAAAAA:lpddziVA6z4-wSKRrnBpptwivJ48BQ3Pmxxd5cxJk_nullA3wQoeGF8BkJ7Dx1ek1bI4DvgK_9KpeYw
>
> [6] Q Vuong et al. Supervised policy update for deep reinforcement learning. https://arxiv.org/abs/1805.11706
>
> [7] HF Song et al. V-mpo: On-policy maximum a posteriori policy optimization for discrete and continuous control. https://arxiv.org/abs/1909.12238
>
> [8] S Huang et al. Cleanrl: High-quality single-file implementations of deep reinforcement learning algorithms. https://github.com/vwxyzjn/cleanrl

---

> > ### Comment · Reviewer_HKZk · 2024-11-22
> > **Response to Authors**
> >
> > Thank you for the detailed responses and revisions. I believe that the proposed objective in (14) is an interesting and potentially valuable contribution, provided that it is clearly presented, positioned appropriately with respect to prior work, and supported by comprehensive experimental evidence. Unfortunately, my main concerns have not been addressed, and I do not think that the current version of the paper is suitable for publication.
> >
> > - **Theoretical novelty of Section 3:** The results in this section are not novel and do not acknowledge connections to prior work. The formulation of (9)-(10) uses results previously shown in [1], and motivates the optimization problem in (15) that is almost identical to the $L^\infty$ constraint formulation in [6] ([6] adds an extra constraint). Proposition 3.2 / Theorem 3.3 directly follow from Pinsker’s inequality which has been used throughout the policy improvement literature (including the original TRPO paper as acknowledged in lines 145-160). I also do not believe that Proposition 3.2 / Theorem 3.3 add much value to the paper, since the authors impose constraints in (15) that restrict the feasible set to be smaller than the TV divergence trust region. The paper could go directly from (10) to (15), and then use (15) to motivate (14).
> > - **Presentation of Section 4:** Section 4 does not clearly present and explain the important characteristics of the SPO objective that result in experimental benefits compared to PPO. The strongly convex quadratic structure with its unique optimal solution seems to be useful / important, but this is not clear from the class of $\epsilon$-aligned functions introduced in Definition 4.1. Based on the assumptions in Definition 4.1, I think it is possible to write an objective with the same set of optimal points as PPO that satisfies Definition 4.1: $rA - |A| |r – r^*|$. Meanwhile, there are simple objectives such as $(r-r^*)^2$ that are convex with a unique optimal solution of $r^*$, but do not satisfy Definition 4.1.
> > - **Related work:** Thank you for adding a Related Work section, but the treatment of prior work should be more thorough and should not be deferred to the Appendix. I believe that it is important to discuss related work and clearly position your contribution relative to this work in the main paper.
> > - **Experiments:** Thank you for providing additional experimental results. The new comparison with adaptive learning rates is promising. I would suggest focusing the paper more heavily on the experimental benefits of SPO which I believe is the main contribution. Comparisons to well-tuned versions of PPO for each architecture size would strengthen the experimental analysis. Non-parametric methods such as [6] are probably the most similar to the ideas proposed in this paper and should also be compared against (for the same target probability ratio, the only difference is the choice of objective function used for the parametric projection).

---

> > > ### Author Response · Authors · 2024-11-23
> > >
> > > Dear Reviewer HKZk,
> > >
> > > Thank you for your comments. We will address your concerns below.
> > >
> > > >**The results in this section are not novel and do not acknowledge connections to prior work.**
> > >
> > > The derivation from (9) to (10), although we believe that this is a widely used result, we have added a citation to [1] on line 222 in the current version of our paper.
> > >
> > > >**I also do not believe that Proposition 3.2 / Theorem 3.3 add much value to the paper.**
> > >
> > > We think Propositions 3.2 and Theorem 3.3 are necessary because the performance improvement lower bound
> > > $$\mathcal{L}=E_{s\sim{\rho_{\pi}(\cdot)},a\sim{\pi(\cdot|s)}}\left[\frac{\tilde{\pi}(a|s)}{\pi(a|s)}\cdot A_{\pi}(s,a)\right]-C\cdot D_{\mathrm{TV}}^{\max}(\pi,\tilde{\pi})^2$$
> > > **cannot be directly optimized** due to the unknown $C=\frac{4\epsilon\gamma}{(1-\gamma)^2}$ and the maximum value of the TV divergence.
> > >
> > > So, the commonly used approach is to attempt to constrain the latter term $D_{\mathrm{TV}}^{\max}(\pi,\tilde{\pi})^2$ while optimizing the former term (surrogate objective) to **indirectly** optimize the lower bound $\mathcal{L}$. Therefore, in this **indirect optimization case**, we need Proposition 3.2 and Theorem 3.3 to show that the TV divergence-constrained space is still a better choice for optimizing $\mathcal{L}$. We hope this addresses your concerns.
> > >
> > > >**The paper could go directly from (10) to (14), and then use (14) to motivate (15).**
> > >
> > > Thanks for your suggestion. We sincerely believe that Proposition 3.2 explains why we should constrain the TV divergence rather than the looser KL divergence, as the KL divergence leads to a **smaller** solution space according to Proposition 3.2. Theorem 3.3 further demonstrates that this smaller solution space may result in **missing the truly optimal solution**.
> > >
> > > Therefore, your suggestion of going directly from (10) to (14) does not contradict our approach. However, before that, we need to choose between (4) and (10) (KL dirvergence or TV divergence). We use Proposition 3.2 and Theorem 3.3 to demonstrate that constraining the TV divergence is a better choice, and that's why we choose (10) rather than (4), and then derive (14). We sincerely hope this addresses your concerns.
> > >
> > > >**Based on the assumptions in Definition 4.1, I think it is possible to write an objective with the same set of optimal points as PPO that satisfies Definition 4.1:** $rA-|A||r-r^*|$
> > >
> > > Thank you for your question. However, we believe this is **not correct** because
> > > $$rA-|A|\cdot|r-r^*|=rA-|A|\cdot|r-(1+\mathrm{sign}(A)\cdot\epsilon)|$$
> > > if $A>0$ and $r>1+\epsilon$, we have
> > > $$rA-|A|\cdot|r-(1+\mathrm{sign}(A)\cdot\epsilon)|=rA-A\cdot|r-(1+\epsilon)|=rA-A\cdot(r-(1+\epsilon))=A\cdot(1+\epsilon)$$
> > > which does **not satisfy the Definition 4.1**, as it does not include the $rA$ term.
> > >
> > > >**Meanwhile, there are simple objectives such as** $|r-r^*|$ **that are convex with a unique optimal solution of** $r^*$, **but do not satisfy Definition 4.1.**
> > >
> > > We need to emphasize that including the $rA$ term is necessary because it ensures the gradient is directed towards optimizing the surrogate objective, rather than simply making $r$ approach the constraint boundary $r^*=1+\mathrm{sign}(A)\cdot\epsilon$. We would like to gently remind you that when updating the policy network, a **batch** of data is typically used. If we only optimize $|r-r^*|$ or $(r-r^*)^2$, this would result in all data having **equal advantage weights** and would not optimize the surrogate objective $E_{s\sim{\rho_{\pi}(\cdot)},a\sim{\pi(\cdot|s)}}\left[\frac{\tilde{\pi}(a|s)}{\pi(a|s)}\cdot A_{\pi}(s,a)\right]$.

---

> > > ### Author Response · Authors · 2024-11-23
> > >
> > > >**I believe that it is important to discuss related work and clearly position your contribution relative to this work in the main paper.**
> > >
> > > We have added references to **line 222 [1]** and **line 267 [2]** of the current paper, and we believe that our contributions have been clearly stated. We would like to refer to the comments made by reviewer **ESWe** and **Dgb4** on this paper:
> > >
> > > - **Reviewer EsWe:** _This paper is well-written and clearly structured. The theoretical foundation is straightforward (in a positive way), and the implementation is simple, which could enhance reproducibility._
> > >
> > > - **Review Dgb4:** _The paper is clearly written and provides strong theoretical grounding. New theorems and proofs substantiate the methodology, ensuring that the approach is technically sound._
> > >
> > > This indicates that the other reviewers have a good understanding of the paper and acknowledge our contributions.
> > >
> > > >**Non-parametric methods such as [2] are probably the most similar to the ideas proposed in this paper and should also be compared against.**
> > >
> > > According to your request, we have added experiments with SPU [2], as shown in Figure 4, showing that SPO still achieves the best performance. Overall, we have corrected the presentation order in Section 3 and cited relevant papers, we first introduce the constrained optimization problem (14) and then present our objective (15). We believe that this can help readers understand our paper in a simpler way. We have also demonstrated the necessity of Proposition 3.2 and Theorem 3.3. Unfortunately, due to space limitations, we are unable to include all the related works in the main text.
> > >
> > > **Finally, we would like to thank you for the time you have dedicated to this work. We have done our best to address all of your comments. We also kindly ask you to consider stronger support for the paper if your concerns have been addressed.**
> > >
> > > # References
> > >
> > > [1] J Queeney et al. Generalized Proximal Policy Optimization with Sample Reuse. https://proceedings.neurips.cc/paper_files/paper/2021/hash/63c4b1baf3b4460fa9936b1a20919bec-Abstract.html
> > >
> > > [2] Q Vuong et al. Supervised policy update for deep reinforcement learning. https://arxiv.org/abs/1805.11706

---

> ### Author Response · Authors · 2024-11-26
>
> Dear Reviewer HKZk,
>
> We would like to kindly remind you that, perhaps due to your busy schedule, further feedback on our paper might have been overlooked. Based on your recent comments, we have made the following updates to our paper:
>
> - We have added relevant works [1, 2] in line 222 and line 267.
> - We have corrected the order of presentation in Section 3, first introducing problem (14) and then deriving objective (15).
> - We have clarified the necessity of Proposition 3.2 and Theorem 3.3, noting that the performance improvement lower bound $\mathcal{L}=E_{s\sim{\rho_{\pi}(\cdot)},a\sim{\pi(\cdot|s)}}\left[\frac{\tilde{\pi}(a|s)}{\pi(a|s)}\cdot A_{\pi}(s,a)\right]-C\cdot D_{\mathrm{TV}}^{\max}(\pi,\tilde{\pi})^2$ cannot be directly optimized.
> - We have included comparative experiments with the non-parametric method SPU [2] in Figure 4.
> - We have also addressed the unreasonableness of the two objectives you mentioned: $rA-|A|\cdot|r-r^*|$ and $|r-r^*|$.
> - Due to space limitations, we have included the necessary related work in Appendix A.
>
> Overall, we believe that your concerns have been addressed. If this is the case, we sincerely hope that you can provide stronger support for our work (we noticed that you gave our work the lowest rating with the highest confidence).
>
> We deeply appreciate the time and effort you have invested in reviewing our work. We have made revisions based on your feedback. If such efforts are not acknowledged, it may hinder constructive dialogue in the future review process.
>
> Thank you for your understanding and consideration.
>
> Sincerely,
>
> Authors of Paper #3422
>
> # References
>
> [1] J Queeney et al. Generalized Proximal Policy Optimization with Sample Reuse. https://proceedings.neurips.cc/paper_files/paper/2021/hash/63c4b1baf3b4460fa9936b1a20919bec-Abstract.html
>
> [2] Q Vuong et al. Supervised policy update for deep reinforcement learning. https://arxiv.org/abs/1805.11706

---

> > ### Comment · Reviewer_HKZk · 2024-11-26
> > **Response to Authors**
> >
> > Thank you for the additional responses and revisions. I still have concerns about the results and presentation of Sections 3 and 4.
> >
> > - Proposition 3.2 / Theorem 3.3 do not provide insights about the optimization problem that is actually considered in this work. The optimization problem in (14) considers a **strict subset** of the TV divergence trust region $\Omega_{\textnormal{TV}}$, and the KL divergence trust region $\Omega_{\textnormal{KL}}$ is **not** a subset of the trust region in (14).
> > - The example functions were meant to highlight the lack of precision / clarity in Section 4 and Definition 4.1 in particular. **1)** The function $rA - |A| |r – r^*|$ certainly seems to satisfy the imprecise statement "$f$ involves $rA$ term" (which was the point of the example). **2)** The assumption that $g$ obtains its maximum value at $r^*$ makes sure that the surrogate objective is maximized at every data point in (14) (because $r^*$ is the optimal solution to (17)), so an assumption related to $rA$ is not needed to maximize the surrogate objective. **3)** Theorem 4.2 only holds if $r^*$ is the unique optimal solution to $g$, and it is unclear if this follows from the imprecise assumptions in Definition 4.1.
> >
> > I appreciate the authors’ efforts during the discussion period, and the revised version of the paper is an improvement over the original submission. However, I still do not believe the paper is suitable for publication due to the issues described above. I also believe that the original submission should have been prepared much more carefully, rather than requiring significant revisions during the discussion period.

---

> ### Author Response · Authors · 2024-11-26
>
> Thank you for your response. Here are the issues you mentioned:
>
> >**Proposition 3.2 / Theorem 3.3 do not provide insights about the optimization problem that is actually considered in this work. The optimization problem in (14) considers a strict subset of the TV divergence trust region $\Omega_{\mathrm{TV}}$, and the KL divergence trust region $\Omega_{\mathrm{KL}}$ is not a subset of the trust region in (14).**
>
> Thank you for your suggestion. We would like to reference TRPO [1]:
>
> _"While it is motivated by the theory, this problem is impractical to solve due to the large number of constraints. Instead, we can use a heuristic approximation which considers the average KL divergence."_
>
> TRPO is one of the **most important** papers in the field of model-free reinforcement learning, widely recognized by the academic community. However, even so, **there remains a gap between theory and practical algorithms**. In TRPO, the maximum KL divergence is **relaxed to** the average KL divergence, but this did not prevent the reviewers from accepting the paper. Therefore, we believe this suggests that there is **always a gap** between theoretical results and practical algorithms. It seems that the reviewer expects the gap between our theory and the practical algorithm to be infinitesimally small, which is clearly not achievable.
>
> >**The function** $f=rA-|A|\cdot|r-r^*|$ **certainly seems to satisfy the imprecise statement** "$f$ **involves** $rA$" **term (which was the point of the example).**
>
> Thank you for your suggestion. However, we believe this is **not correct**. If that were the case, we could also consider $f=rA-rA=0$ as satisfying this condition, which would lead to the absurd conclusion that "$f$ involves everything". We believe this is a complete misinterpretation of Definition 4.1. Furthermore, we also tried the objective you proposed, $f=rA-|A|\cdot|r-r^*|$, and found its performance to be very **limited**, and it **fails to constrain the probability ratio**, you can verify this result by running the code we provided. We hope this addresses your concerns.
>
> >**I also believe that the original submission should have been prepared much more carefully, rather than requiring significant revisions during the discussion period.**
>
> Thank you for your suggestions. However, changes to the paper during the review process are inevitable and can help improve the quality of the paper. Meanwhile, the additional experiments in the main text are unavoidable and are primarily meant to address your concerns. Specifically,
>
> - SPU [2] (Figure 4)
> - Figure 6
> - the entire Section 5.3
> - the Related Work in Appendix A.
>
> We believe that these additional experimental results may have addressed most of the reviewers' concerns.
>
> **So far, we have been working hard to address all the issues you raised. However, we have noticed that you have now raised new questions, which we believe may indicate that most of your previous concerns have been resolved. Moreover, regarding your latest questions, we believe that some of them may be based on misunderstandings (see the text above). Additionally, we think that some of these questions pertain to common challenges in the field of model-free reinforcement learning, where inevitable gaps often exist between theoretical concepts and practically feasible algorithms.**
>
> **Thank you again for your time and effort.**
>
> # References
>
> [1] J Schulman et al. Trust Region Policy Optimization. https://people.engr.tamu.edu/guni/csce642/files/trpo.pdf
>
> [2] Q Vuong et al. Supervised policy update for deep reinforcement learning. https://arxiv.org/abs/1805.11706

---

> > ### Comment · Reviewer_HKZk · 2024-11-26
> > **Response to Authors**
> >
> > My initial review contained detailed comments due to significant concerns with the original submission, which largely ignored related work that has been done on the topic. My follow-up comments have been in response to points raised by the authors in their replies, where I have provided detailed clarifications related to these concerns.

---

> ### Author Response · Authors · 2024-11-27
>
> We sincerely appreciate the effort you have put into reviewing our work. It seems inevitable to make revisions during the rebuttal phase, which helps authors improve the quality of their papers. We believe that it might be a better choice for other reviewers and the area chair to make the decision.

---

### Official Review · Reviewer_Xchv · 2024-11-04

**Soundness:** 3
**Presentation:** 3
**Contribution:** 2
**Rating:** 6
**Confidence:** 3

**Summary:**

--Why:
It has been shown that consistent policy improvement can be achieved through constrained policy optimization or trust region policy optimization.

--What:
However, in practice, current methods use approximations to enforce the trust region. In particular, PPO uses clipping of the probability ratio to enforce the trust region constraint. It's simple and works well in practice. However, the author argues and shows that theoretically, this cannot always satisfy the bound. When the probability ratio is out of bounds, this method zeros the gradient, and therefore there isn't any corrective gradient to push the policy back towards the trust region. Thus, this paper investigates better trust region enforcement for PPO.

--How:
Authors propose an objective (eq 15), which directly constrains the total variation divergence instead of KL divergence. They show theoretically that this is a better divergence (see Proposition 3.2 and 3proposition 3.2 .3). They also argue that, in practice, their objective can apply a corrective gradient when the probability ratio goes over the bound (see Figure 2 and Figure 4).

**Strengths:**

Paper is well written, problem statement is clear and is easy to follow. Motivations are supported with references and theoretical justifications as well as toy examples.

**Weaknesses:**

--Originality and Significance

the idea of using  total variation divergence is not new and it's been proposed in original TRPO paper already as authors has also mentioned. Also some follow up work on constraint optimization and safe RL has explored similar ideas that worth being mentioned, for example see:

https://arxiv.org/pdf/1705.10528
,https://arxiv.org/pdf/1805.07708

Given that the idea of using total variation divergence is not entirely new, this paper requires strong experimental evidence to support its claims. The authors' decision to use hyperparameters from the original PPO paper seems unfair, considering that changes in the setup necessitate retuning. However, even with this caveat, Figures 9 and the continuous control results show no significant improvement over the original PPO. I believe tuning PPO could achieve better results. Furthermore, the results in Figure 6 are puzzling. A larger ResNet benefits SPO but not PPO. Observing the constant bias in Figure 7, it appears that choosing a different epsilon for PPO might lead to better results. This suggests that the performance difference could be attributed to suboptimal hyperparameter selection for PPO.

**Questions:**

--What is the significance of proposition 3.2 and theorem 3.3?
Could not we use the theorem (3) to conclude that Total Variation divergence is the better choice compare to KL divergence?


--As the authors rightly mentioned, larger ResNets or deeper MLPs could help better representation learning and, therefore, better performance, as shown in Figures 5 and 6. I have personally observed this to be true in my own experiments with PPO. Could the authors provide a more compelling reason for why PPO in their experiments does not benefit from better representation learning, beyond the claim that PPO's performance deteriorates with an increasing number of parameters? Could tuning hyperparameters help PPO in this case?

--In figure 7, is it possible to match the curve for SPO(epsilon=0.2) if we choose different epsilon for PPO?

---

> ### Author Response · Authors · 2024-11-19
>
> Dear Reviewer Xchv,
>
> Thank you for your valuable feedback, which will greatly help improve our work! Below, we will address your concerns and look forward to further interaction with you.
>
> To address your concerns, we have submitted all the **code for PPO and SPO used in this paper** as supplementary material. This code is primarily based on the high-quality reinforcement learning library CleanRL [1]. Additionally, a **new version of the paper** has been uploaded, with the main revisions **highlighted in red** for your convenience in further review.
>
> # I. The Novelty of Using TV Divergence Constraints
>
> >**the idea of using total variation divergence is not new**
>
> >**What is the significance of proposition 3.2 and theorem 3.3? Could not we use the theorem (3) to conclude that Total Variation divergence is the better choice compare to KL divergence?**
>
> We acknowledge that the concept of Total Variation (TV) divergence is indeed not new and has been previously explored in various contexts. However, our application of TV divergence in this work presents **several unique contributions**:
>
> - Firstly, we formally prove that optimizing the lower bound with a TV divergence constraint is better than using the KL divergence-constrained lower bound. Next, we will explain **why this conclusion cannot be directly derived from Theorem (3)**. According to (3), the performance improvement lower bound is
> $$\eta(\tilde{\pi})-\eta(\pi)\geq L_{\pi}(\tilde{\pi})-L_{\pi}(\pi)-C\cdot D_{\mathrm{TV}}^{\max}(\pi,\tilde{\pi})^2$$
> where $$C=\frac{4\epsilon\gamma}{(1-\gamma)^2},\enspace\epsilon=\max_{s,a}|A_{\pi}(s,a)|,\enspace L_{\pi}(\tilde{\pi})=\eta(\pi)+E_{s\sim{\rho_{\pi}(\cdot)},a\sim{\tilde{\pi}(\cdot|s)}}\left[A_{\pi}(s,a)\right]$$
> Denote this lower bound as
> $$\mathcal{L}=L_{\pi}(\tilde{\pi})-L_{\pi}(\pi)-C\cdot D_{\mathrm{TV}}^{\max}(\pi,\tilde{\pi})^2$$
> which we aims to optimize. However, note that this lower bound $\mathcal{L}$ **can not be directly optimized** due to the **unknown coefficient** $C$ and the **maximum operator**. Otherwise we could directly obtain the conclusion that TV divergence is the better choice compare to KL divergence, as TV divergence lower bound is tighter. However, one fact is that we optimize the lower bound $\mathcal{L}$ **indirectly** by maximizing
> $$L_{\pi}(\tilde{\pi})-L_{\pi}(\pi)=E_{s\sim{\rho_{\pi}(\cdot)},a\sim{\pi(\cdot|s)}}\left[\frac{\tilde{\pi}(a|s)}{\pi(a|s)}\cdot A_{\pi}(s,a)\right]$$
> while trying to constrain the TV divergence $D_{\mathrm{TV}}^{\max}(\pi,\tilde{\pi})^2$. Therefore, we need to use **Proposition 3.2** and **Theorem 3.3** to demonstrate that, in this **indirect optimization case**, the TV divergence is still the better choice. In other words, there are cases where the TV divergence is large, but the $L_{\pi}(\tilde{\pi})-L_{\pi}(\pi)$ is **much larger**. Therefore, even if the TV divergence is not bounded, this is still a good lower bound. However, this situation is not within the scope of discussion for commonly used policy improvement algorithms due to the unknown coefficient $C$ mentioned earlier. We hope this can address your concern.
>
> - Secondly, although previous work has made efforts to constrain the TV divergence, our work provides **strong theoretical guarantees** for constraining the probability ratio. This is because our new objective
> $$f_{\mathrm{spo}}=rA-\frac{|A|}{2\epsilon}\cdot(r-1)^2,\enspace r=\frac{\tilde{\pi}(a|s)}{\pi(a|s)}$$
> has been proven to be convex with respect to the probability ratio $r$, and the maximum is achieved exactly **at the constraint boundary** $r=1+\mathrm{sign}(A)\epsilon$. Meanwhile, constraining the probability ratio indirectly constrains the average Total Variation (TV) divergence under the training dataset, which is
> $$D_{\mathrm{TV}}^{\max}(\pi,\tilde{\pi})^2=\frac{1}{4}\max_{s}\left[{E}_{a\sim\pi(\cdot|s)}\left|\frac{\tilde{\pi}(a|s)}{\pi(a|s)}-1\right|\right]^2$$
> Since the **max operator needs to consider all possible states**, it is generally relaxed by using heuristic approximations to replace the max operation with an average operation (the expectation). Nevertheless, our objective involves imposing probability ratio constraints on **each** data point $(s_t,a_t)$, which results in **stricter constraints** compared to heuristic approximations and aligns **more closely** with the max operator in the lower bound $\mathcal{L}$.
>
> - Lastly, we would like to quote **reviewer ESWe and reviewer Dgb4's comments** related to the use of TV divergence constraints in our paper: (**ESWe**) _Although several papers have already pointed out issues with the heuristic clipping technique in PPO and proposed alternatives, I find the authors' approach offers a novel perspective, leading me to lean towards acceptance._ (**Dgb4**) _The paper is clearly written and provides strong theoretical grounding. New theorems and proofs substantiate the methodology, ensuring that the approach is technically sound._

---

> ### Author Response · Authors · 2024-11-19
>
> # II. Why does PPO Perform Worse with Deeper Policy Networks?
>
> > **The authors' decision to use hyperparameters from the original PPO paper seems unfair, considering that changes in the setup necessitate retuning.**
>
> > **Could the authors provide a more compelling reason for why PPO in their experiments does not benefit from better representation learning, beyond the claim that PPO's performance deteriorates with an increasing number of parameters? Could tuning hyperparameters help PPO in this case?**
>
> > **In figure 7, is it possible to match the curve for SPO(epsilon=0.2) if we choose different epsilon for PPO?**
>
> Thank you for your thoughtful review and the insightful questions! We would like to provide a more detailed explanation regarding why PPO in our experiments does not seem to benefit from better representation learning, and whether tuning hyperparameters could help in this case.
>
> - **Performance Decline with Increased Parameters.** Firstly, we provide an explanation for this phenomenon in the main text, suggesting that the clipping operation in PPO can result in the gradients of some data points being zero. A potential issue is that these data points **can still affect the performace improvement lower bound** $$\mathcal{L}=L_{\pi}(\tilde{\pi})-L_{\pi}(\pi)-C\cdot D_{\mathrm{TV}}^{\max}(\pi,\tilde{\pi})^2=E_{s\sim{\rho_{\pi}(\cdot)},a\sim{\pi(\cdot|s)}}\left[\frac{\tilde{\pi}(a|s)}{\pi(a|s)}\cdot A_{\pi}(s,a)\right]-\frac{C}{4}\max_{s}\left[{E}_{a\sim\pi(\cdot|s)}\left|\frac{\tilde{\pi}(a|s)}{\pi(a|s)}-1\right|\right]^2$$
> even if they produce zero gradients, because all the current data come from the sampling of the old policy, and none of them can be ignored. Therefore, the clipping operation used by PPO is likely to **cause the latter term in the lower bound to become very large**. This is because these zero-gradient data points can be influenced by the gradients produced by non-zero gradient data points during further training, as the parameters of the policy are **constantly changing**. Optimizing the first term in the lower bound will inevitably push the current policy further away from the old policy. This situation is common during PPO training. While deeper networks have better representation capabilities, they are also **more sensitive to parameter changes** (with a larger Lipschitz constant). Therefore, even though deeper networks can allow for better representation learning, if the lower bound of policy improvement is **poor or even negative**, it can lead to worse performance.
>
>
> - **Hyperparameter Tuning.** We have added more experiments to support our conclusions, including **selecting different hyperparameters for PPO (Figure 6)** and **different $\epsilon$ (Figure 7)**, as well as further comparisons with other methods that restrict the magnitude of policy updates **(Figure 8)**. These results indicate that naively reducing epsilon in the clipping objective does not overcome the issue of ratio deviation. Additionally, the comparative experiments between SPO and PPO with different hyperparameters and network depths indeed confirm the superiority and robustness of SPO's performance.
>
> [1] S Huang et al. Cleanrl: High-quality single-file implementations of deep reinforcement learning algorithms. https://github.com/vwxyzjn/cleanrl

---

> ### Author Response · Authors · 2024-11-23
>
> Dear Reviewer Xchv,
>
> We have addressed your concerns and provided additional experimental results. What do you think of our response as well as new results? We also kindly ask you to consider stronger support for the paper if your concerns have been addressed.
>
> Thank you very much.
>
> Sincerely,
> The Authors of Paper #3422

---

> > ### Comment · Reviewer_Xchv · 2024-11-28
> >
> > Thank you for the detailed response. My questions are all answered so I adjust my score accordingly.

---

> > > ### Author Response · Authors · 2024-11-29
> > >
> > > Dear Reviewer Xchv,
> > >
> > > we sincerely appreciate your constructive feedback and inspiring support for our work.

---

### Author Response · Authors · 2024-11-21
**Happy to Respond to Any Further Comments/Clarifications**

To All Reviewers,

Thank you for the effort you put into reading and commenting on our work! If you have any questions, please feel free to discuss them with us. We look forward to further interactions with you.

Thanks for your time and support!

Authors of Paper #3422

---

### Note · Authors · 2025-01-23

I have read and agree with the venue's withdrawal policy on behalf of myself and my co-authors.